# Lightweight Image-to-3D Shape Generation via Vitality-Aware Pruning and Quantization

## Abstract

We propose the *first* compression framework for image-to-3D generative models that substantially reduces model size while preserving synthesis fidelity. Recent advances in 3D shape generative modeling, particularly Diffusion Transformer (DiT) architectures, have achieved remarkable progress in synthesis fidelity and controllability. However, the substantial computational cost of large DiT-based image-to-3D models hinders their practical application in resource-constrained settings. While existing efficiency-oriented approaches improve inference speed, they leave the underlying model size and computational cost of synthesis largely unchanged. To address this challenge, we propose a systematic compression framework that physically reduces model size while preserving the fidelity of 3D shape synthesis. Our approach builds on the observation that Transformer layers in 3D DiT models exhibit non-uniform importance, with only a subset of layers contributing significantly to geometry generation. Leveraging this insight, we introduce a vitality-guided framework that integrates structured pruning, adaptive quantization, and targeted fine-tuning to balance efficiency and quality. Experimental results show that our method achieves up to **66%** model-size reduction across state-of-the-art 3D generative models with minimal loss in synthesis fidelity. This highlights the potential of our framework as a plug-and-play solution for efficient 3D shape generation across diverse models.

## 1 Introduction

The growing demand for high-quality 3D content has driven the evolution of generative models beyond early VAE (Chen et al., 2025b), GAN (Gao et al., 2022), and diffusion approaches (Poole et al., 2023) toward more advanced architectures such as Diffusion Transformers (DiT) (Peebles & Xie, 2023; Wu et al., 2024b), with flow-based models emerging as a promising alternative (Lipman et al., 2023; Xiang et al., 2025; Zhao et al., 2025). Despite their impressive progress in 3D shape fidelity, current image-to-3D generative models remain computationally demanding, as large DiT architectures incur substantial memory and inference costs that hinder their use in cost-sensitive or real-time applications. Although several studies (Tochilkin et al., 2024; Lai et al., 2025) have explored improving the efficiency of 3D generation, they primarily focus on inference acceleration and neglect model-size reduction, resulting in limited impact on overall computational requirements. To overcome these constraints, we propose a novel compression framework that directly reduces model complexity while maintaining the fidelity of shape synthesis.

Recent investigations into transformer-based diffusion models have shown critical insights about layer-wise contribution patterns across different generation tasks. Studies in text-to-image (Avrahami et al., 2025) and text-to-video (Kim et al., 2025) synthesis demonstrate that only specific layers significantly influence the quality of final outputs, while others contribute minimally to the generation process. These findings have been successfully leveraged for text-based editing applications, enabling targeted modifications of existing foundation models without additional training processes. Extending this principle to the 3D generation domain, we show that DiT layers in image-to-3D models exhibit similar importance patterns and introduce a vitality metric that quantifies each layer's contribution to shape synthesis.

Importantly, we use this analysis to develop a systematic compression framework for existing foundation models, specifically targeting the DiT component responsible for the denoising process in

image-to-3D generation. While prior methods have sought to optimize or distill entire pipelines (Wu et al., 2024b; Lai et al., 2025), we deliberately preserve the encoder–decoder and rendering components, focusing instead on the computationally dominant DiT stage. This design choice is motivated by both practical and scientific considerations, as the DiT governs multi-view reasoning and geometric consistency that are essential for high-quality shape synthesis, making it the most critical target for compression. Unlike methods designed to improve efficiency in video or 2D generative models (An et al., 2023; Fang et al., 2023), where redundancy primarily arises from spatial or temporal correlations, compression in 3D DiTs must additionally preserve geometric coherence across views and depth.

Guided by our layer-wise vitality analysis, we first prune layers whose vitality scores fall below a threshold, removing redundant computation while preserving core functionality. We then apply adaptive quantization to the remaining layers, allocating higher precision to critical layers and more aggressive compression to less vital ones. Finally, we perform targeted finetuning to systematically recover performance degradation introduced by compression. Together, these steps achieve substantial model-size reduction while maintaining the generative fidelity essential for high-quality 3D shape synthesis.

Our experimental results demonstrate that our approach successfully achieves substantial model compression while preserving synthesis quality across multiple state-of-the-art models, including Step1X-3D (Li et al., 2025) ($-65.63\%$), Hunyuan3D 2.0 ($-66.37\%$), and Hunyuan3D 2mini (Zhao et al., 2025) ($-44.50\%$). To the best of our knowledge, we are the ***first*** to systematically reduce both the parameter count and bit-width of the denoising transformer in an image-to-3D shape generative model, achieving substantial model-size compression while preserving 3D geometric fidelity. We expect to expand our framework into a generalized, plug-and-play solution that enables high-quality 3D shape synthesis across diverse existing frameworks under limited computational resources.

To summarize, we introduce the following contributions:

- We present an analysis of layer-wise contributions in Diffusion Transformers (DiTs) for image-to-3D shape generation and introduce a vitality computation method tailored for 3D tasks.
- Building on this analysis, we propose the *first* model-size reduction approach that incorporates layer vitality into a unified pruning and adaptive quantization.
- We introduce an efficient finetuning strategy that targets only low-vitality layers, effectively restoring performance with minimal additional cost.
- We demonstrate our method on three DiT-based models, obtaining significantly smaller networks that maintain performance comparable to their full-sized counterparts.

## 2 RELATED WORK

**3D Generative Models.** 3D generative models have evolved across various representations, including voxels (Wu et al., 2016; Xie et al., 2020; Mittal et al., 2022), point clouds (Luo & Hu, 2021; Zhou et al., 2021; Vahdat et al., 2022), implicit fields (Zheng et al., 2022; Hui et al., 2022; Shue et al., 2023; Chou et al., 2023), and meshes (Nash et al., 2020; Siddiqui et al., 2024). Early GAN-based approaches such as EG3D (Chan et al., 2022) and pi-GAN (Chan et al., 2021) demonstrated promising view-consistent synthesis but were constrained by limited category diversity and training data. Diffusion-based models later improved geometric fidelity, with Shape-E (Jun & Nichol, 2023) introducing one of the first text-to-3D diffusion frameworks and inspiring subsequent methods that jointly model geometry and appearance. More recently, large-scale 3D datasets such as Objaverse (Deitke et al., 2023) have enabled powerful Large Reconstruction Models (LRMs) (Hong et al., 2024; Tang et al., 2024; Zhang et al., 2024a; Tochilkin et al., 2024; Liu et al., 2023b; Xu et al., 2024) for single-pass 3D synthesis, while next-generation systems including 3DTopia-XL (Chen et al., 2025c) and GaussianAnything (Lan et al., 2025) leverage triplane-based and scalable Gaussian representations for high-quality open-domain generation. However, these existing models often produce coarse geometries that require memory-intensive refinement. To address these limitations, recent methods adopt a two-stage pipeline combining compact geometry generation with multi-view diffusion for texturing (Zhang et al., 2024b; Li et al., 2025; Zhao et al., 2025), while others explore Structured Latent (SLAT) representations (Xiang et al., 2025). Despite these advances, substantial memory and computational demands remain a key obstacle to the widespread adoption of 3D generative modeling.

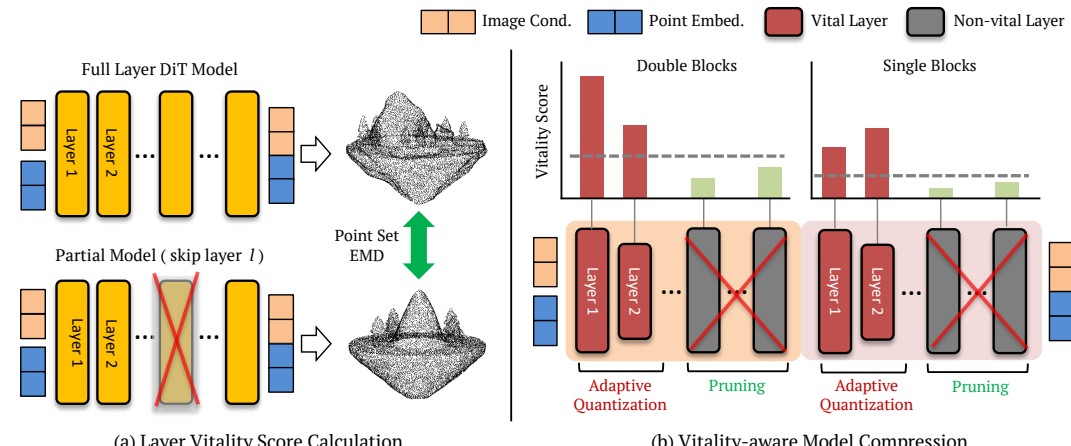

Figure 1: **Method Overview. (a)** Method of layer vitality score calculation. In order to analyze the contribution of individual layer $l$ from DiT model, we calculate the point cloud distance between full layer model output and layer ablated model output. In our case, we use Earth Mover's Distance (EMD). **(b)** Based on the calculated vitality scores, we remove the redundant layers which show low vitality scores. In this step, we apply different thresholds to double-block and single-block layers.

**Model Compression for Transformer-based Models.** While recent 3D generative models (Li et al., 2025; Zhao et al., 2025; Xiang et al., 2025) have achieved remarkable improvements in fidelity, they still suffer from extremely high memory consumption. Although methods like Turbo3D (Hu et al., 2025) and FlashVDM (Lai et al., 2025) attempt to address efficiency, they mainly focus on accelerating inference rather than fundamental model compression. In the broader Transformer literature, prior work have shown various pruning approaches, including attention head, block, and layer pruning (Fan et al., 2020; Lee et al., 2024; Fang et al., 2025), can effectively reduce model complexity while maintaining performance. Extensive research have explored quantization, spanning from low-bit BERT models (Zafrir et al., 2019; Shen et al., 2020) to recent DiT-specific schemes (Wu et al., 2024a; Chen et al., 2025a; Hwang et al., 2025). These methods consistently demonstrate that substantial memory savings can be achieved without compromising generation quality. In addition, knowledge distillation techniques (Sanh et al., 2019; Jiao et al., 2019; Wang et al., 2020) have proven effective in recovering accuracy after compression. Despite these advances, 3D generative modeling lacks a systemic investigation into Transformer layer vitality and its application to pruning and quantization, which forms the central motivate of our work.

## 3 METHOD

Our primary objective is to physically reduce the model size of 3D shape generation DiT architectures. To achieve this, we first quantitatively analyze the contribution of each Transformer layer to the final output (Sec. 3.1). This analysis allows us to identify and prune redundant layer whose importance is negligible, thereby improving efficiency. Subsequently, we apply adaptive quantization guided by analyzed vitality, constructing a lightweight model that almost preserves the performance of original model (Sec. 3.2). To further reduce the performance degradation, we finetune the compressed model to closely match the accuracy of the full model (Sec. 3.3).

### 3.1 VITALITY ANALYSIS OF 3D DiT LAYERS

We begin by measuring the contribution of each layer in the 3D DiT model to the final output. In prior work (Avrahami et al., 2025) on T2I generative models, the vitality of a layer is evaluated by comparing the outputs of the full DiT framework with that of a model where a target single layer $l$ is removed. The perceptual difference between the two outputs is measured using the DINO (Caron et al., 2021) distance, and layers that induce larger discrepancies regarded as more important.

Following a similar principle, we analyze the Image-to-3D DiT layers using layer ablation in Fig. 1 (a). Given the same conditional input image $y$, we generate a point set using the full model $\theta_{\text{full}}$ and

layer-ablated model $\theta_{-l}$ by removing $l$-th layer. The distance between these point sets then serves as a quantitative indicator of vitality. Since perceptual distance used in the image domain cannot be applied directly, here we require a metric suitable for 3D point sets. We therefore adopt Earth Mover's Distance (EMD) to measure the vitality of 3D DiT layers, as it effectively captures overall geometric differences between point sets.

For a conditional image $y$, our vitality score is defined as:

$$\text{vitality}(l) = \mathbb{E}_{y \sim \mathcal{D}} \left[ \min_{\Gamma \in \mathcal{P}_n} \frac{1}{n} \sum_{i=1}^{n} \sum_{j=1}^{n} \Gamma_{ij} \left\| q_{\theta_{\text{full}}}^{(i)}(y) - q_{\theta_{-l}}^{(j)}(y) \right\|_2 \right], \tag{1}$$

where $\mathcal{D}$ is an image dataset, $n$ denotes the number of points in each point cloud, $q_{\theta_{\text{full}}}(y)$ is point cloud generated from full model, $q_{\theta_{-l}}(y)$ is point cloud generated from layer $l$ removed model, and permutation matrices are defined as $\mathcal{P}_n = \left\{ \Gamma \in \{0,1\}^{n \times n} \ \middle| \ \sum_{j=1}^{n} \Gamma_{ij} = 1, \ \sum_{i=1}^{n} \Gamma_{ij} = 1, \ \forall i, j \right\}$.

In contrast to the Chamfer Distance, which computes nearest neighbor correspondences and mainly reflects local geometric accuracy, EMD computes the optimal transport cost between two point sets, producing a one-to-one correspondence that accounts for the overall distribution of the shape. This property enables EMD to detect global structural distortions such as shifts, asymmetry, or large-scale misalignment that may occur when a layer responsible for maintaining geometric coherence is removed. Consequently, EMD provides a more faithful measure of a layer's functional contribution to preserving overall structural integrity beyond local surface similarity. Moreover, since EMD formulates the comparison as a mass transport problem, it is less biased toward dense or unevenly sampled surface regions, ensuring consistent and fair vitality evaluation across shapes of varying mesh density. To further support the robustness of the proposed evaluation metric, we present a quantitative comparison in App. C.1. For completeness, we also report the corresponding analysis using the Chamfer Distance in App. E.1, which exhibits a consistent overall trend and further validates the reliability of our EMD-based evaluation.

Figure 2 (a) shows the results of our analysis on the Step1X-3D (Li et al., 2025) model, computed from 210 randomly generated images by DALL·E 3 (Betker et al., 2023) using text prompts from Objaverse (Deitke et al., 2023). Interestingly, most layers are found to have vitality scores that converge close to zero, indicating negligible importance. This pattern is consistent across both of single- and double-block layers. Similar trends are observed in other image-to-3D generation models, including Hunyuan3D 2.0 and Hunyuan3D 2mini (Zhao et al., 2025) (see App. E), through with slightly weaker magnitudes.

The qualitative analysis in Fig. 2 (b) make this effect more tangible. Skipping vital double-block layers produces severe geometric distortions, such as unintended rotations, while removing vital single-block layers leads to degrade finer details and artifacts. Conversely, omitting low-vitality layers in either cases barely effects the output.

## 3.2 MODEL COMPRESSION USING VITAL LAYERS

**Layer Pruning.** Based on the vitality scores, we determine which layers to prune using a threshold $\tau$. Layers with vitality scores exceeding $\tau$ are classified as vital and retained, while the rest are pruned. However, we observe that applying a single threshold across both double- and single-block layers cause performance degradation. To mitigate this, we introduce separate thresholds, $\tau_d$ and $\tau_s$, for double- and single-block layers, respectively. To determine these thresholds, we progressively remove layers starting from the lowest vitality score and monitor the distance to the vanilla model output. The threshold is chosen at the point where a sharp drop in quality occurs. We provide the detailed selection process in App. C.

**Adaptive Quantization.** After pruning, we further reduce the model size through quantization. Here, we also leverage the vitality scores to assign different bit-widths to each layer. To minimize performance loss while maximizing compression, we define two groups: highly vital layers are quantized to 8-bit, and less-vital layers to 4-bit. Similar to pruning, distinct thresholds are applied to double-block and single-block layers to avoid performance drops. Since our method primarily focuses on layer-wise analysis, we apply weight-only quantization and do not consider activations.

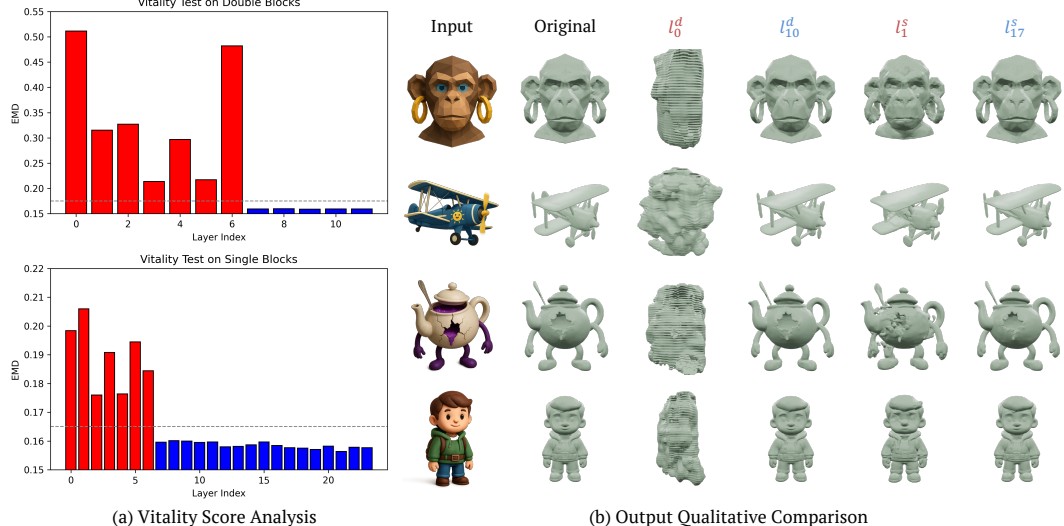

Figure 2: **Layer Vitality Analysis.** **(a)** Vitality score analysis on Step1X-3D model. Layers with red index are vital layers which has high contribution to synthesis, whereas most layers appear in blue, indicating negligible contribution. **(b)** Qualitative analysis of layer vitality on Step1X-3D. Removing vital layers noticeable degradation in shape outputs, while removing non-vital layers yields only minor differences. Note that $l_i^d$ indicates the $i$-th double-block layer and $l_j^s$ denotes the $j$-th single-block layer, with indexing starting from 0.

## 3.3 DISTILLATION FINE-TUNING

While our proposed pruning and quantization yield an efficient compression, the resulting model may not fully replicate the behavior of the full model. To bridge this gap, we perform finetuning so that the compressed model better follows the dynamics of the full model as shown in Fig. 3. Unlike standard flow matching training, our approach focuses on maximizing similarity between the compressed and full models. Specifically, we design a loss function to encourage the student to imitate the ODE path of the full model such as:

$$\mathcal{L}_{\text{Distill}}(\theta_c) = \frac{1}{2} \left\| v^c(z_t^f, t, y) - v^f(z_t^f, t, y) \right\|_2^2 + \frac{1}{2} \left\| v^c(z_t^f, t, \varnothing) - v^f(z_t^f, t, \varnothing) \right\|_2^2, \quad (2)$$

where $v^c$ is model prediction output from compressed model $\theta_c$, $v^f$ is output from full model $\theta_{full}$, $z_t^f$ is latent of timestep $t$ sampled from full model, $y$ is input image condition, and $\varnothing$ is null condition. In order to obtain more accurate distillation, we calculate distances for both of conditional and unconditional model predictions. For each individual timestep, we optimize the parameters of weights from compressed model. After single optimization step at timestep $t$, we jump into next step $t - 1$ using flow sampling with full-model prediction output.

However, finetuning all remaining vital layers is computationally inefficient and, in some cases, causes the compressed student model to diverge further from the full teacher model, leading to degraded performance. To mitigate this, we propose a selective finetuning strategy. Specifically, we choose the vital layer with the lowest vital score (denoted as "Min-vital" in Fig. 3) and finetune only its weights, thereby avoiding excessive modification of vital layers.

## 4 EXPERIMENT

**Experimental Details.** To validate our proposed method, we conduct experiments on three Image-to-3D shape generation models. We use the state-of-the-art models Step1X-3D (Li et al., 2025), Hunyuan3D 2.0 and 2mini (Zhao et al., 2025). As described in Sec. 3.2, based on results of the vitality analysis, we set the standard for eliminating redundant layers and for setting thresholds to determine 8-bit and 4-bit layers. For example, for Step1X-3D, we apply $\tau_d = 0.17$ for double-block layers and $\tau_s = 0.165$ for single-block layers, and set thresholds of 0.25 and 0.185 for double-block

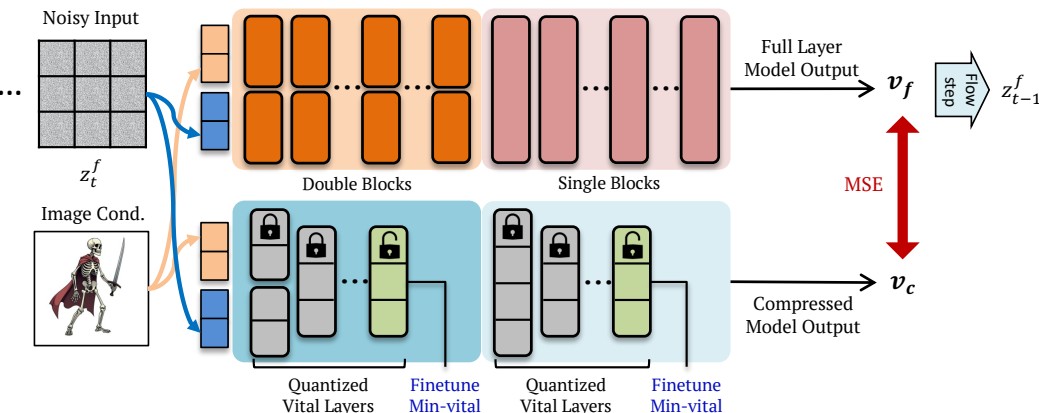

Figure 3: **Targeted Finetuning Pipeline.** To refine the compressed model, we finetune the minimally vital (Min-vital) layer of compressed model so that the model output closely matches that the full-layer model. Specifically, during the full-layer model flow sampling path, we optimize the compressed student model to reproduce the full model's output under the same condition and latent input.

and single-block layers to determine the 8-bit and 4-bit layers. During the non-vital layer finetuning stage, we use rendered images from subset 10K of Objaverse (Deitke et al., 2023) dataset. For Step1X-3D, we train with a learning rate of $10^{-8}$, and for Hunyuan3D 2.0 and 2mini, we used $10^{-4}$. In both cases, we conduct finetuning process for 30K iterations for Step1X-3D, and 20k iterations for Hunyuan3D models. For sampling, we use timestep of 30 for Step1X-3D and 20 for Hunyuan3D models. We provide more experimental details in App. A.

**Evaluation Metrics.** For evaluation, we employ two embedding-based metrics that measure semantic correspondence between input images and generated 3D meshes: **Uni3D-I** (Zhou et al., 2024) and **OpenShape-I** (Liu et al., 2023a). Both models compute similarity in a joint image–3D embedding space, providing an objective measure of alignment quality. We report results on 200 image–shape pairs sampled from Objaverse (Deitke et al., 2023). For validation, we generate 200 images using DALL·E 3 (Betker et al., 2023) from text prompts originally provided by Objaverse (Deitke et al., 2023).

In addition, we measure the model size, specifically the memory footprint of its parameters, to evaluate spatial efficiency after compression. Furthermore, we evaluate geometric consistency during compression using the **volume (V-IoU)** and **symmetric surface IoU (SS-IoU)** scores with rigid alignment, as shown in App. D.2.

**Baselines.** We compare our method with a diverse set of 3D generation approaches, spanning feedforward, diffusion, and transformer-based paradigms:

- **Splatter Image** (Szymanowicz et al., 2024): a diffusion-based model that progressively generates 3D from images, achieving higher realism but often struggling with fine-grained alignment.

- **TripoSR** (Tochilkin et al., 2024): a fast feedforward model that directly predicts 3D shapes from images, designed for lightweight inference but with limited geometric fidelity.

- **LGM** (Tang et al., 2024): a Gaussian-based feedforward approach that produces compact 3D representations, prioritizing efficiency over detailed reconstruction.

- **Craftsman3D** (Li et al., 2024): a transformer-based DiT model with strong mesh generation quality, though requiring large memory and computation.

- **TRELLIS** (Xiang et al., 2025): another state-of-the-art DiT-based architecture that excels in generating structured 3D meshes, but comes with significant model size overhead.

Table 1: **Overall Quantitative Results. (a)** Quantitative comparison with baselines, including original 3D generative models. Compared with the scores of the original frameworks and other 3D generative models, our approach successfully maintains high synthesis performance under compression. **(b)** User study results. Our compression strategy preserves perceptual quality, achieving performance nearly indistinguishable from the full model.

(a) Quantitative comparison with baselines.

| Models | Metrics | | |
|---|---|---|---|
| | Uni3D-I ↑ | OpenShape-I ↑ | Size (GB) ↓ |
| Splatter Img | 0.1800 | 0.0681 | 0.661 |
| TripoSR | 0.2994 | 0.1313 | 0.622 |
| LGM | 0.2482 | 0.1108 | 0.800 |
| Craftsman3D | 0.3519 | 0.1455 | 2.322 |
| TRELLIS | 0.3442 | 0.1455 | 2.175 |
| Step1X-3D | 0.3586 | 0.1480 | 2.452 |
| **Step1X-3D + Ours** | 0.3580 | 0.1489 | 0.843 |
| Hy3D 2.0 | 0.3582 | 0.1487 | 2.704 |
| **Hy3D 2.0 + Ours** | 0.3601 | 0.1491 | 0.909 |
| Hy3D 2mini | 0.3614 | 0.1490 | 1.042 |
| **Hy3D 2mini + Ours** | 0.3608 | 0.1484 | 0.578 |

(b) User study results.

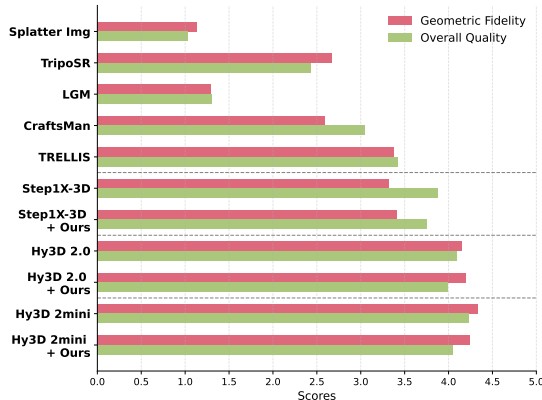

## 4.1 QUANTITATIVE RESULTS

In Tab. 1 (a), we show the quantitative comparison results between our proposed lightweight model and other baselines. As already shown in the previous part, we use same baseline methods including reference models of Step1X-3D , Hunyuan3D 2.0, and Hunyuan3D 2mini. For fair comparison, we only calculate parameter size of backbone models (Unet or Transformer), without considering subsidiary networks such as autoencoder and condition encoders. Comparing with early methods of Splatter Image, TripoSR and LGM, the mesh quality and perceptual scores are largely degraded comparing with our methods although they have relative small model size. With recent models of Craftsman3D and TRELLIS, quantitative scores are higher than other baselines, however they still do not outperform our best model (Hunyuan3D 2mini + Ours), in terms of mesh generation quality and model size.

We also illustrate the comparison results between reference full models and our compressed versions. For larger models of Step1X-3D and Hunyuan3D 2.0, our compressed model can reduce the model size over 50% but still show minor degradation with almost same level of performance. We also apply further compression on already-compressed model of Hunyuan3D 2mini. Surprisingly, our method still can be applied to small model with negligible degradation. Overall results indicate that our proposed compression method successfully reduce the model size while maintaining synthesis quality.

To further access the perceptual quality of our proposed method, we present user study results in Tab. 1 (b). To evaluate the quality of 3D shape generation, participants were asked two questions: (1) whether the correspondence between the image and the generated shape was reasonable (Geometric Fidelity), and (2) whether the quality of the generated 3D mesh was satisfactory (Overall Quality). Details of the user study setup are provided in the App. B.

Consistent with our quantitative results, we observe that earlier works such as Splatter Image, LGM, and TripoSR exhibit substantially lower perceptual mesh quality compared to other models. Recent methods, like Craftsman3D and TRELLIS, show improvements over the earlier models but still fall short of ours. Notably, our compressed frameworks achieve high performance nearly indistinguishable from the full model baseline. This also demonstrates that our compression method effectively preserves the performance of the full model.

## 4.2 QUALITATIVE RESULTS

We qualitatively compare our method with representative baselines across different model families as shown in Fig. 4. Compared to the diffusion-based Splatter Image, which often struggle to capture

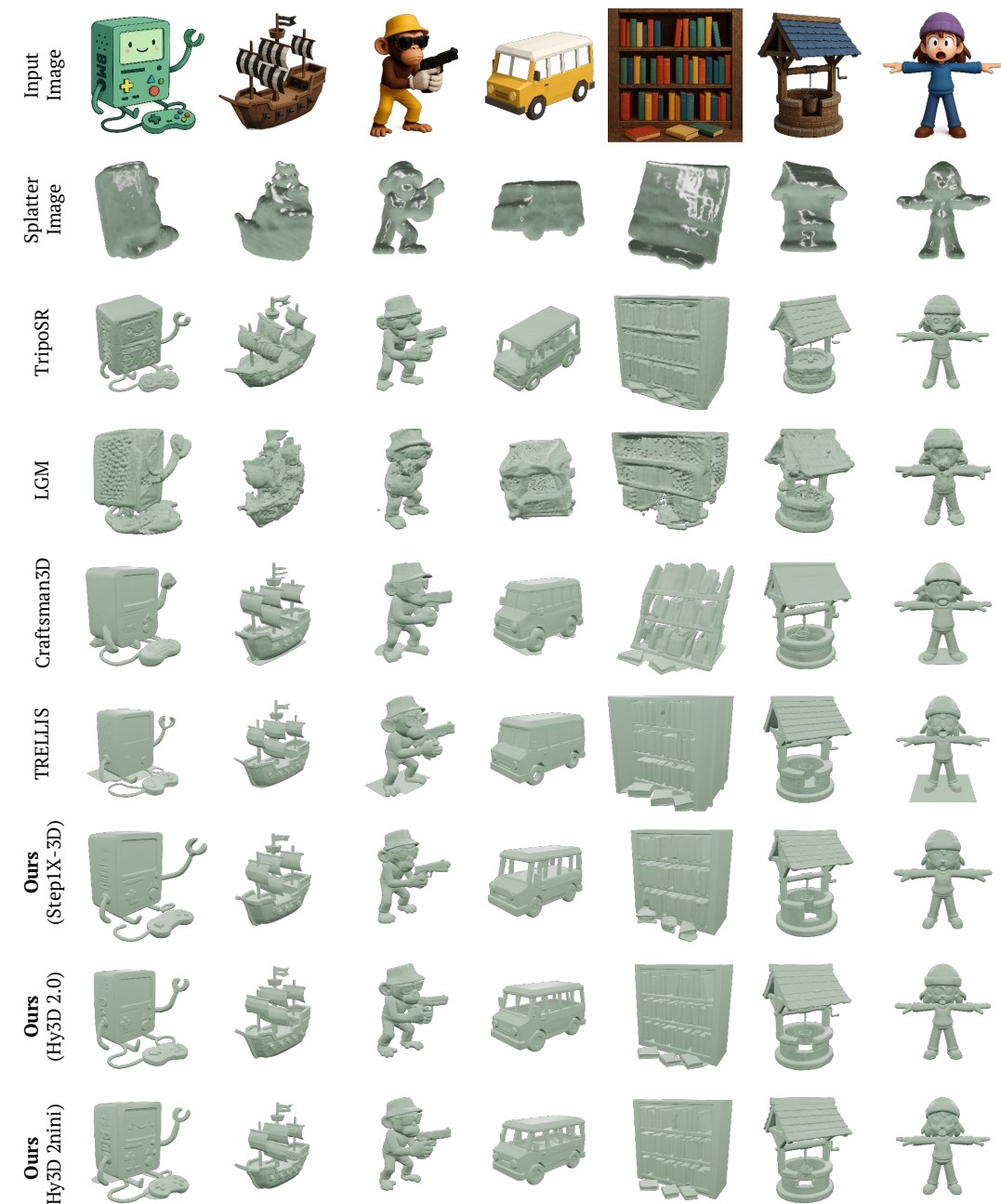

Figure 4: **Qualitative Comparison with Baselines.** For conditional image-to-3D mesh generation, earlier works such as Splatter Image, TripoSR, and LGM often produce meshes with lost details or struggle to match the alignment with the input image. Recent models like Craftsman3D and Trellis achieve good quality but still fall slightly short of ours in terms of fine details. Our models deliver superior perceptual performance while maintaining a significantly reduced model size compared to prior approaches.

fine details or maintain strong alignment with the input image, our approach achieves superior shape generation quality with smaller model sizes. Against feedforward models such as TripoSR and LGM, our method produces more detailed and faithful reconstructions, whereas the baseline often fails to capture fine image-specific features and exhibits artifacts. In addition, compared to recent DiT-based models (Craftsman3D, TRELLIS), our framework generates meshes with sharper details and stronger image–shape correspondence.

Table 2: **Quantitative Comparison on Ablation Study.** To verify the effect of our proposed components, we measure image-3D shape correspondence scores on various experimental settings. We conduct ablation study on both of Step1X-3D and Hunyuan3D models. (**Bold**: best score, Underline: second best, Colored mark : within 1% of the best score.)

| Conditions | Step1X-3D | | | Hunyuan3D 2.0 | | | Hunyuan3D 2mini | | |
|---|---|---|---|---|---|---|---|---|---|
| | Uni3D-I ↑ | OpenShape-I ↑ | Size (GB) ↓ | Uni3D-I ↑ | OpenShape-I ↑ | Size (GB) ↓ | Uni3D-I ↑ | OpenShape-I ↑ | Size (GB) ↓ |
| Original | 0.3586 | 0.1480 | 2.452 | 0.3582 | 0.1487 | 2.704 | **0.3614** | **0.1490** | 1.042 |
| ✚ Pruning (random) | 0.0829 | 0.0375 | 1.123 | 0.1171 | 0.0606 | 1.575 | 0.3084 | 0.1356 | 0.954 |
| ✚ Vitality Analysis | 0.3584 | 0.1472 | 1.123 | 0.3576 | **0.1491** | 1.575 | 0.3437 | 0.1417 | 0.954 |
| ✚ Quantization (4b) | 0.3489 | 0.1466 | **0.803** | 0.3134 | 0.1351 | **0.709** | 0.3356 | 0.1399 | **0.442** |
| Quantization (8b) | **0.3601** | 0.1479 | 0.910 | 0.3574 | 0.1488 | 1.031 | 0.3426 | 0.1420 | 0.622 |
| ✚ Adaptive Quant. | 0.3579 | 0.1478 | 0.843 | 0.3528 | 0.1480 | 0.909 | 0.3437 | 0.1425 | 0.578 |
| ✚ Finetuning (**Ours**) | 0.3580 | **0.1489** | 0.843 (−65.63%) | **0.3601** | **0.1491** | 0.909 (−66.37%) | 0.3608 | 0.1484 | 0.578 (−44.50%) |

## 4.3 ABLATION STUDY

**Quantitative Ablation Study.** For detailed evaluation of our proposed components, we show quantitative measurement in Tab. 2. To evaluate the versatility of our proposed method, we conduct ablation study on 3 different models of Step1X-3D, Hunyuan3D 2.0, and Hunyuan3D 2mini. Starting from the full-parameter original model, we first show the output from random layer pruned model (✚ Pruning (random)). Since many vital layers are removed, the overall quality of model is significantly degraded. Then we apply our vitality-aware pruning strategy, where we prune only non-vital layers (✚ Vitality Analysis). With removing the redundant layers, we can dramatically remove the model size with minimal performance drop. This result clearly show the effectiveness of our proposed pruning stage.

With layer pruned model, we apply quantization to remaining layers (✚ Quantization). With 8bit quantization, we can further reduce the model size, and the performance is slightly degraded or similar to the original model. However, with 4bit quantization, we can see the model size is further decreased but the quality of the model has been dropped, especially for the Hunyuan3D models. With applying our proposed adaptive quantization(✚ Adaptive Quant), we can further reduce the model from 8bit quantization while minimizing the performance drop. After using our finetuning strategy (✚ Finetuning), we are able to achieve performance of the compressed model that was nearly identical to that of the full-parameter model. In the case of Step1X-3D, the difference between the vital and non-vital layers is clear, therefore we can obtain a good model during the pruning step and finetuning had little effect.

**Qualitative Ablation Study.** To clearly demonstrate the effect of each step in our method, we provide qualitative comparisons as shown in Fig. 5. The model with only random pruning applied shows severe degradation. With vitality-aware pruning, performance remains similar to the original, though artifacts appear in the Hunyuan3D models. Under uniform 4-bit quantization, performance drops while quality is partially restored when applying our adaptive quantization. Nevertheless, the Hunyuan3D models still exhibit artifacts. After finetuning, all models achieve results almost identical to those of the full-parameter models.

## 5 CONCLUSION

In this work, we address the challenge of reducing the computational burden of large image-to-3D generative models while maintaining high synthesis quality. We present a vitality-aware compression framework that integrates layer pruning, adaptive quantization, and targeted fine-tuning to systematically reduce model complexity. Through extensive experiments on state-of-the-art architectures, including Step1X-3D, Hunyuan3D 2.0, and Hunyuan3D 2mini, our approach achieves over 50% reduction in model size with minimal degradation in 3D shape fidelity. These results highlight that analyzing layer vitality effectively identifies structural redundancies within DiT architectures, enabling substantial compression while avoiding performance degradation in 3D shape synthesis.

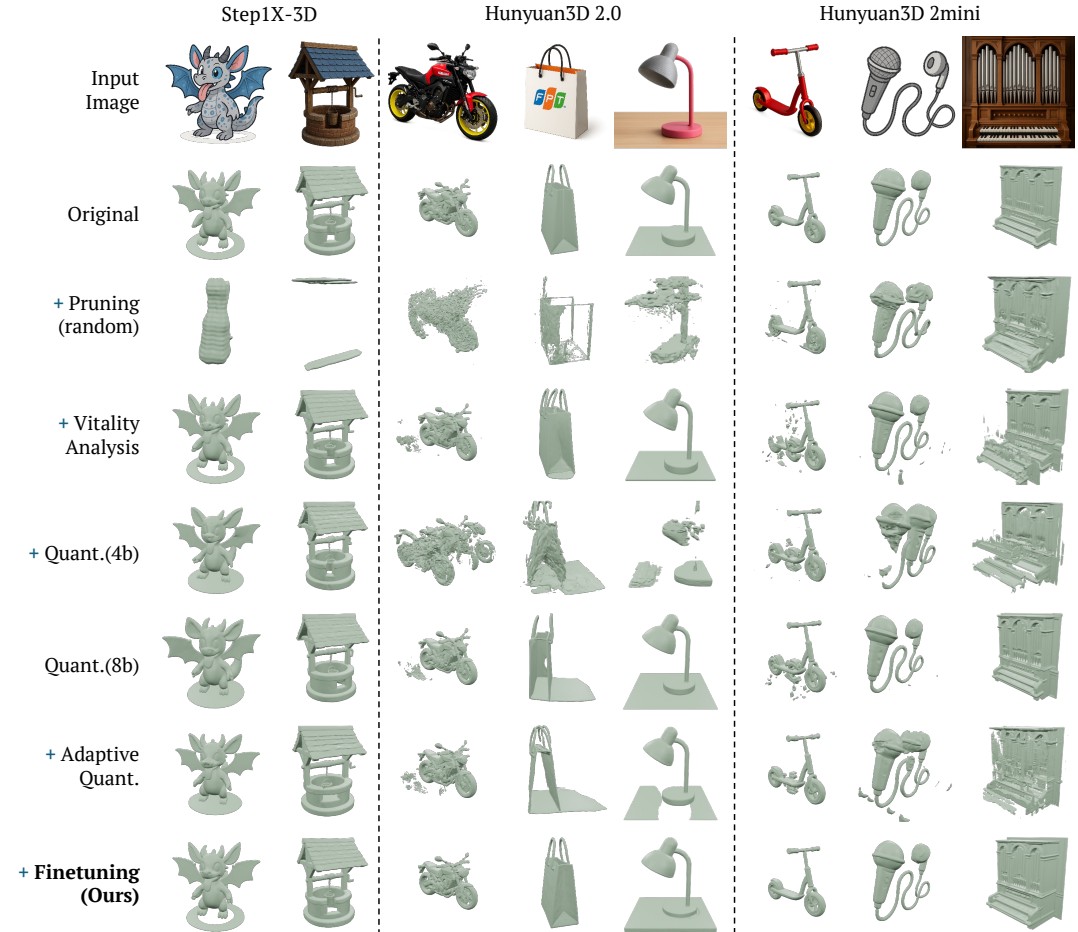

Figure 5: **Qualitative Comparisons on Ablation Study.** With random pruning, the model suffers from severe mesh degradation. In contrast, pruning only non-vital layers yields results nearly identical to the original. Applying 4-bit quantization causes noticeable detail loss, especially in the Hunyuan models. Adaptive quantization attains quality comparable to 8-bit while further reducing size. Finally, combined with our finetuning, the compressed model achieves results almost indistinguishable from the original.

Our framework, as the first approach for physical model compression of 3D shape generative models, opens up new possibilities for scalable, plug-and-play 3D generation in resource-constrained and interactive environments.

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

# APPENDIX

In this appendix, we provide additional experimental details (App. A), user study settings (App. B), and supplementary methodological explanations (App. C). We also include extended results for both baseline comparisons (App. D.1) and ablation studies (App. D.2), a detailed analysis of vitality layers (App. E), as well as discussions on limitations and future directions (App. F). Finally, we describe the use of large language models (LLMs) throughout our workflow (App. G).

## A  ADDITIONAL EXPERIMENTAL DETAILS

For Hunyuan3D 2.0, we set $\tau_d = 0.18$ and $\tau_s = 0.17$ for layer pruning in double-block and single-block DiT, respectively, and apply thresholds of 0.21 and 0.16 for adaptive quantization of double-block and single-block layers. Meanwhile, since we observe that every double-block layer in Hunyuan3D 2mini plays a significant role in shape generation (Fig. E), we do not apply layer pruning and set all layers to 8-bit in quantization except for layer 4. For the single-block layers in the same model, we set $\tau_s = 0.192$ to remove redundancy, and apply thresholds of 0.2 for single-block layers, respectively, to determine whether a layer should be assigned higher (8-bit) or lower (4-bit) bits during adaptive quantization.

For each model, the indices of the target layers (with indexing starting from 0) are as follows: Step1X-3D has target layers at index 3 for the double-block and 2 for the single-block. Hunyuan3D 2.0 has target layers at index 11 for the double-block and 26 for the single-block. Hunyuan3D 2Mini has target layers at index 4 for the double-block and 12 for the single-block.

Furthermore, we conduct model compression experiments under the following training settings: Step1X-3D is trained for 22 hours on 2 A100 GPUs with a batch size of 10 per GPU; Hunyuan3D 2.0 requires 50 hours on 2 A100 GPUs with a batch size of 3 per GPU; and Hunyuan3D 2mini is trained for 14 hours on a single A200 GPU with a batch size of 20.

## B  USER STUDY DETAILS

We conducted a user study involving 31 participants. For each question, six different input image setups were presented, and the participants were asked to assign a score from 1 (low) to 5 (high). Each question included the mesh output of the original model subject to compression, along with the results of other baselines, as described in Tab. 1(b), which were randomly shuffled before being attached to the survey. The evaluation questions consist of:

- **Geometric fidelity**: on a scale from 1 to 5, rate how reasonable the generated shape represents the overall geometry of the object in the input image.
- **Overall synthesis quality**: evaluate each generated 3D shape on a 1–5 scale, where 5 indicates highest synthesis quality and 1 indicates the lowest.

## C  METHODOLOGICAL DETAILS

### C.1  COMPARISONS ON ROBUSTNESS OF VITALITY METRICS

We validate the robustness of our vitality-aware metrics using the double-layer DiT block from the Step1X-3D model (Li et al., 2025), with 210 images used for vitality analysis. To assess stability across sampling densities, we vary the number of points extracted from the meshes (5k, 10k, and 15k) and report the resulting Chamfer Distance (CD) and Earth Mover's Distance (EMD) in Tab. A. We observe that deeper layers (*e.g.*, layers 7–11) are more sensitive to sampling density, with CD values changing significantly as the sampling density varies. This instability arises from CD's dependence on nearest-neighbor correspondences, which makes it sensitive to sampling density and spatial distribution.

In contrast, EMD remains comparatively stable, with differences no greater than 5% relative to our main results (measured with 10k points), even when using only 5k sampled points. This indicates that EMD provides a more stable measure of geometry correspondence under varying sampling conditions.

Table A: **Quantitative Comparison for Robustness of Vitality Metrics on Step1X-3D.** Comparison of Chamfer Distance (CD) and Earth Mover's Distance (EMD) across training scales (5k, 10k, 15k samples) on the double-layer DiT block of the Step1X-3D  CD diff and EMD diff denote percentage deviations from the 10k baseline. Note that all differences are reported in absolute values.

| # index | 10k Points | | 15k Points | | | | 5k Points | | | |
|---|---|---|---|---|---|---|---|---|---|---|
| | CD | EMD | CD | EMD | CD diff (%) | EMD diff (%) | CD | EMD | CD diff (%) | EMD diff (%) |
| 0 | 0.1641 | 0.5116 | 0.1720 | 0.5159 | **4.82** | **0.82** | 0.1711 | 0.5078 | **4.28** | **0.75** |
| 1 | 0.0628 | 0.3152 | 0.0759 | 0.3294 | **20.92** | **4.50** | 0.0790 | 0.3253 | **25.87** | **3.21** |
| 2 | 0.0613 | 0.3270 | 0.0633 | 0.3414 | **3.31** | **4.38** | 0.0646 | 0.3281 | **5.46** | **0.31** |
| 3 | 0.0160 | 0.2136 | 0.0134 | 0.2064 | **16.06** | **3.36** | 0.0134 | 0.2055 | **16.22** | **3.81** |
| 4 | 0.0404 | 0.2970 | 0.0436 | 0.3037 | **7.88** | **2.28** | 0.0401 | 0.3015 | **0.74** | **1.53** |
| 5 | 0.0138 | 0.2170 | 0.0179 | 0.2149 | **29.60** | **0.94** | 0.0159 | 0.2127 | **14.99** | **1.97** |
| 6 | 0.1183 | 0.4822 | 0.1244 | 0.5047 | **5.18** | **4.66** | 0.1218 | 0.4833 | **3.01** | **0.23** |
| 7 | 0.0012 | 0.1591 | 0.0007 | 0.1554 | **40.62** | **2.30** | 0.0013 | 0.1566 | **12.16** | **1.57** |
| 8 | 0.0014 | 0.1597 | 0.0008 | 0.1594 | **40.22** | **0.16** | 0.0014 | 0.1530 | **2.67** | **4.17** |
| 9 | 0.0010 | 0.1588 | 0.0006 | 0.1559 | **43.83** | **1.83** | 0.0012 | 0.1547 | **16.44** | **2.59** |
| 10 | 0.0009 | 0.1591 | 0.0004 | 0.1556 | **51.70** | **2.23** | 0.0011 | 0.1568 | **30.77** | **1.46** |
| 11 | 0.0010 | 0.1595 | 0.0005 | 0.1575 | **50.03** | **1.24** | 0.0012 | 0.1551 | **16.17** | **2.73** |

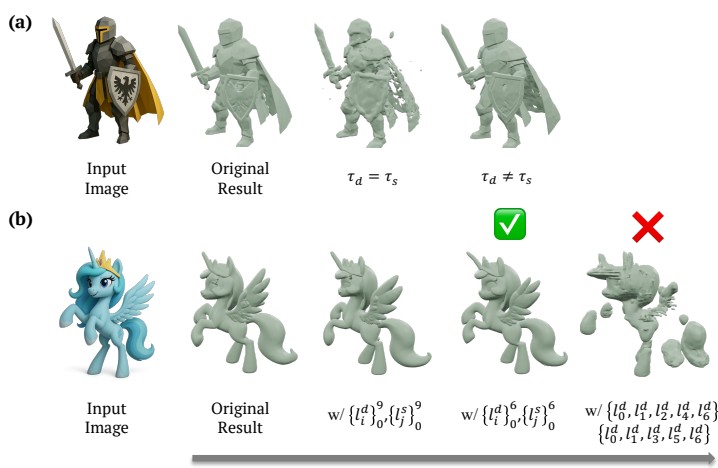

Figure A: **Details of Layer Pruning Process.** **(a)** Applying identical pruning criteria to both double- and single-block layers in Hunyuan3D 2.0 causes geometric distortion. We therefore use distinct thresholds for the two layer types to preserve structural fidelity. **(b)** Layer elimination process of Step1X-3D. Minor details change below the threshold, but beyond it, the mesh structure collapses. Below the threshold, only fine details are altered, whereas exceeding it causes the mesh structure to collapse.

Overall, these results demonstrate that the vitality-aware EMD metric remains robust across changes in sampling resolution, preserving consistent behavior at different point densities, whereas CD becomes increasingly unreliable when fewer samples are used.

## C.2    IDENTIFICATION OF NON-VITAL LAYERS FOR PRUNING

Figure A(a) shows a failure case when the same pruning criterion is applied to both double-block and single-block layers. Specifically, we compare our method against a pruning attempt on Hunyuan3D 2.0 using a shared threshold of $\tau_d = \tau_s = 0.18$. The geometry becomes severely distorted when applying the same standard to both layers. Based on this observation, we adopt separate pruning criteria for double- and single-block layers.

Meanwhile, as mentioned in Sec. 3.2, we sequentially eliminate layers beginning with those that have the lowest vitality scores, tracking how the results diverge from the baseline model output. The procedure is illustrated in Fig. A (b). We observe that up to a certain threshold, only minor details are affected while the overall shape remains similar. However, beyond this point, the mesh structure becomes completely distorted.

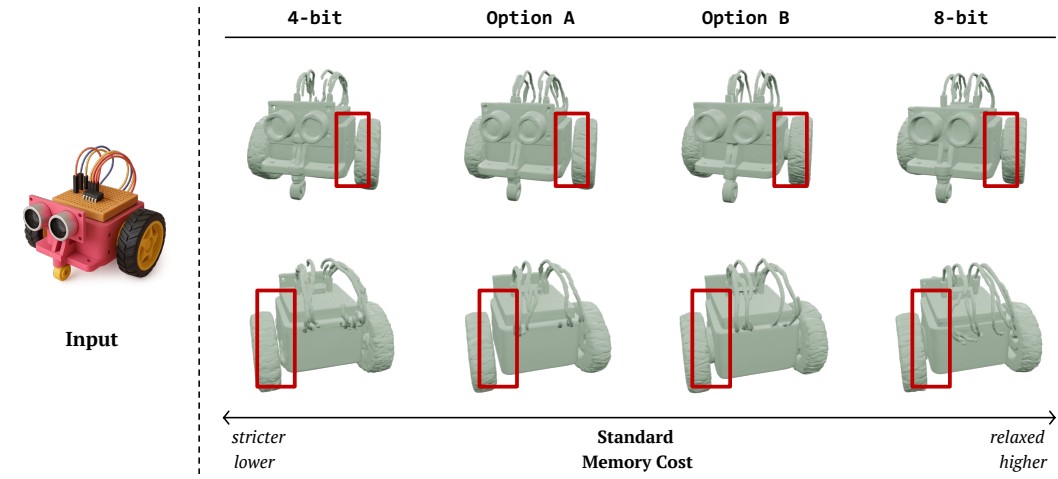

Figure B: **Results of Ablation of Adaptive Quantization Strategies on Step1X-3D.** A stricter quantization setting in adaptive quantization leads to a more degraded initial model state. When comparing the marked regions across the results, a clear synthesis degradation can be observed as stricter quantization criteria are applied. Consequently, achieving higher compression rates at this stage requires more extensive finetuning under the same layer pruning configuration.

## C.3  CRITERIA FOR ADAPTIVE QUANTIZATION

We compare the results before and after finetuning using different adaptive quantization thresholds, as shown in Fig. B. Increasing the strictness of the threshold makes it progressively more difficult to preserve the original model performance. Although the threshold in adaptive quantization can be freely chosen by the user, applying a stricter setting generally requires longer training or more extensive finetuning to maintain stability.

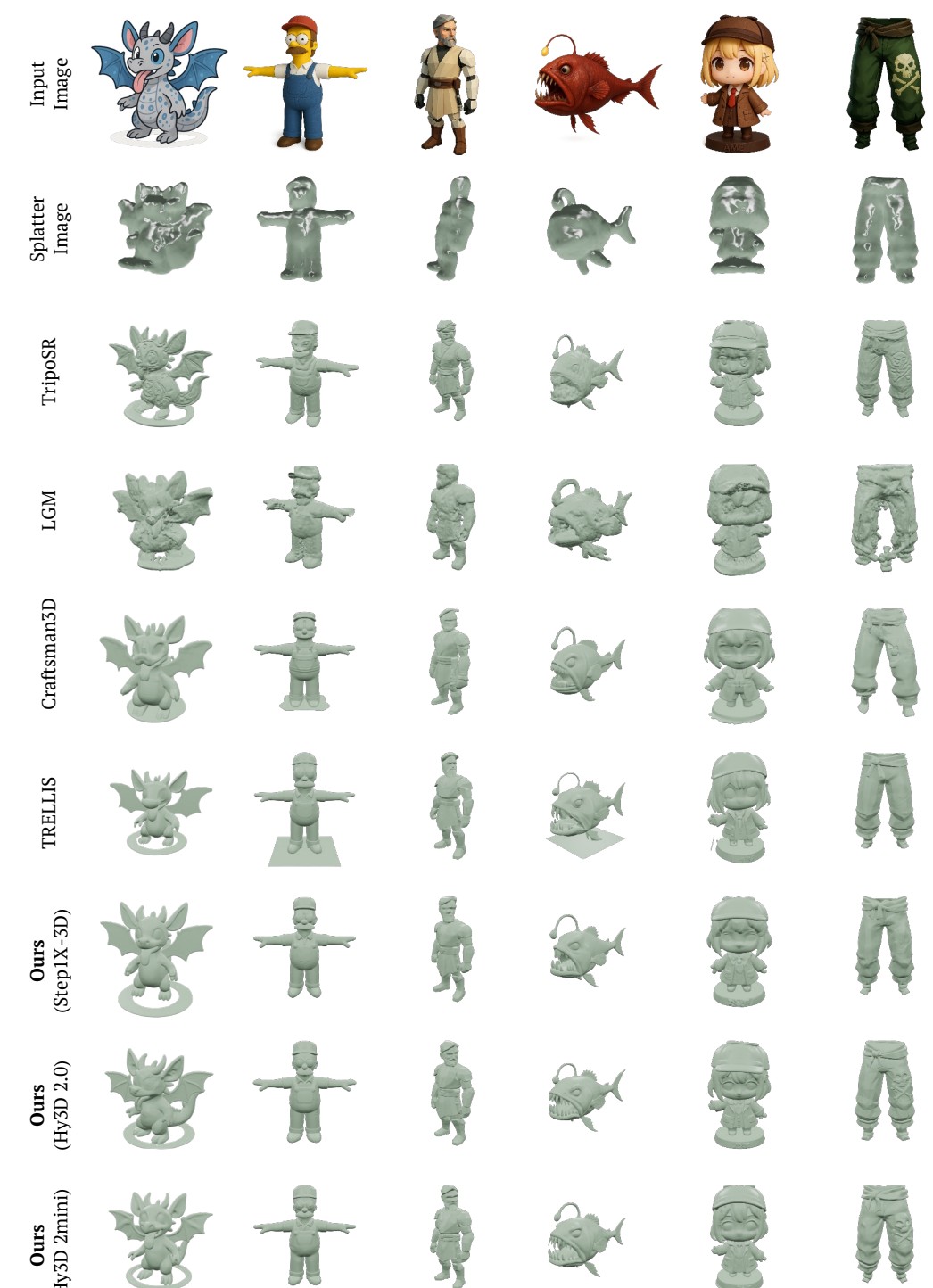

Figure C: **Additional Qualitative Comparison with Baselines.** Our lightweight model generates meshes of higher quality than other baselines, similar to the original model.

Table B: **Additional Quantitative Comparison on Ablation Study.** We evaluate geometric correspondence metrics (V-IoU and SS-IoU) under various ablation settings for both Step1X-3D and Hunyuan3D models. This demonstrates that our approach effectively mitigates performance degradation during compression across diverse DiT-based foundation frameworks.

| Conditions | Step1X-3D | | Hunyuan3D 2.0 | | Hunyuan3D 2mini | |
|---|---|---|---|---|---|---|
| | V-IoU (%) ↑ | SS-IoU (%) ↑ | V-IoU (%) ↑ | SS-IoU (%) ↑ | V-IoU (%) ↑ | SS-IoU (%) ↑ |
| Original | – | – | – | – | – | – |
| + Pruning (random) | 6.01 | 9.16 | 27.50 | 27.94 | 59.53 | 55.73 |
| + Vitality Analysis | 79.27 | 77.29 | 71.32 | 68.66 | 74.08 | 72.05 |
| + Quantization (4b) | 62.56 | 44.69 | 51.49 | 49.49 | 69.40 | 66.34 |
| Quantization (8b) | 69.25 | 67.09 | 69.32 | 66.64 | 73.72 | 71.71 |
| + Adaptive Quant. | 61.11 | 58.60 | 68.06 | 65.21 | 72.66 | 69.71 |
| + Finetuning (**Ours**) | **71.12** | **68.82** | **72.04** | **68.31** | **73.77** | **70.36** |

# D  ADDITIONAL RESULTS

## D.1  QUALITATIVE RESULTS FOR BASELINE COMPARISON

Additional qualitative comparison results can be found in Figure C. This demonstrates that our approach achieves higher performance in 3D shape synthesis compared to existing baselines including recent DiT-based generative models (Li et al., 2024; Xiang et al., 2025), as the original model does.

## D.2  ABLATION STUDY

Table C: **VRAM Allocation and Inference Comparison.** We report VRAM usage and inference time during the denoising process on a single NVIDIA RTX 3090 GPU.

| Model | Variant | VRAM (GB) ↓ | Time (s) ↓ |
|---|---|---|---|
| Step1X-3D | Vanilla | 6.881 | 47.73 |
| | Ours | 3.463 | 18.06 |
| Hunyuan3D 2.0 | Vanilla | 5.187 | 8.34 |
| | Ours | 4.033 | 6.93 |
| Hunyuan3D 2mini | Vanilla | 3.944 | 1.55 |
| | Ours | 3.333 | 1.45 |

**Quantitative Results**  To validate geometric consistency during compression, we additionally provide quantitative ablations using volume and surface IoU metrics measured between the original and compressed models, as shown in Tab. B. Although our compressed models achieve slightly lower performance than those using only vitality-aware layer pruning, considering the exact model size reported in Tab. 2 and the overall quality illustrated in Fig. 5, our method effectively restores synthesis quality while requiring minimal computational overhead.

We also evaluate inference-time savings through our compression process as in Tab. C. Since our approach involves a layer pruning step that dynamically reduces the model size (as shown in Tab. 2), it also improves inference efficiency in both time and memory usage, even though the primary objective is physical model compression. Since Step1X-3D undergoes the most extensive pruning, it achieves the greatest reduction in inference cost. Meanwhile, the Hunyuan models show more moderate improvements since we apply a less aggressive strategy in pruning layers before the subsequent steps. Additional system-level optimization for quantized layers could further improve efficiency in the inference stage.

**Qualitative Results**  Further qualitative ablation results for Hunyuan3D 2.0 and Hunyuan3D 2mini are presented in Fig. D and Fig. E, respectively. In Hunyuan3D models, naive pruning and quantization lead to floaters and collapsed geometry, whereas our compression method produces models that closely match the original in quality.

**Component-Wise Ablations Before Finetuning**  Figure F visualizes the reconstruction quality after applying possible conditions of pruning and quantization on Hunyuan3D 2.0, as well as the model's initial state before finetuning. By comparing the outputs of the vanilla and pruned models across different quantization conditions, we show that our vitality-based pruning approach reduces spatial cost with minimal degradation in synthesis quality. Furthermore, as also shown in Fig. 5, applying 4-bit quantization to all layers causes the model to struggle in forming coherent overall

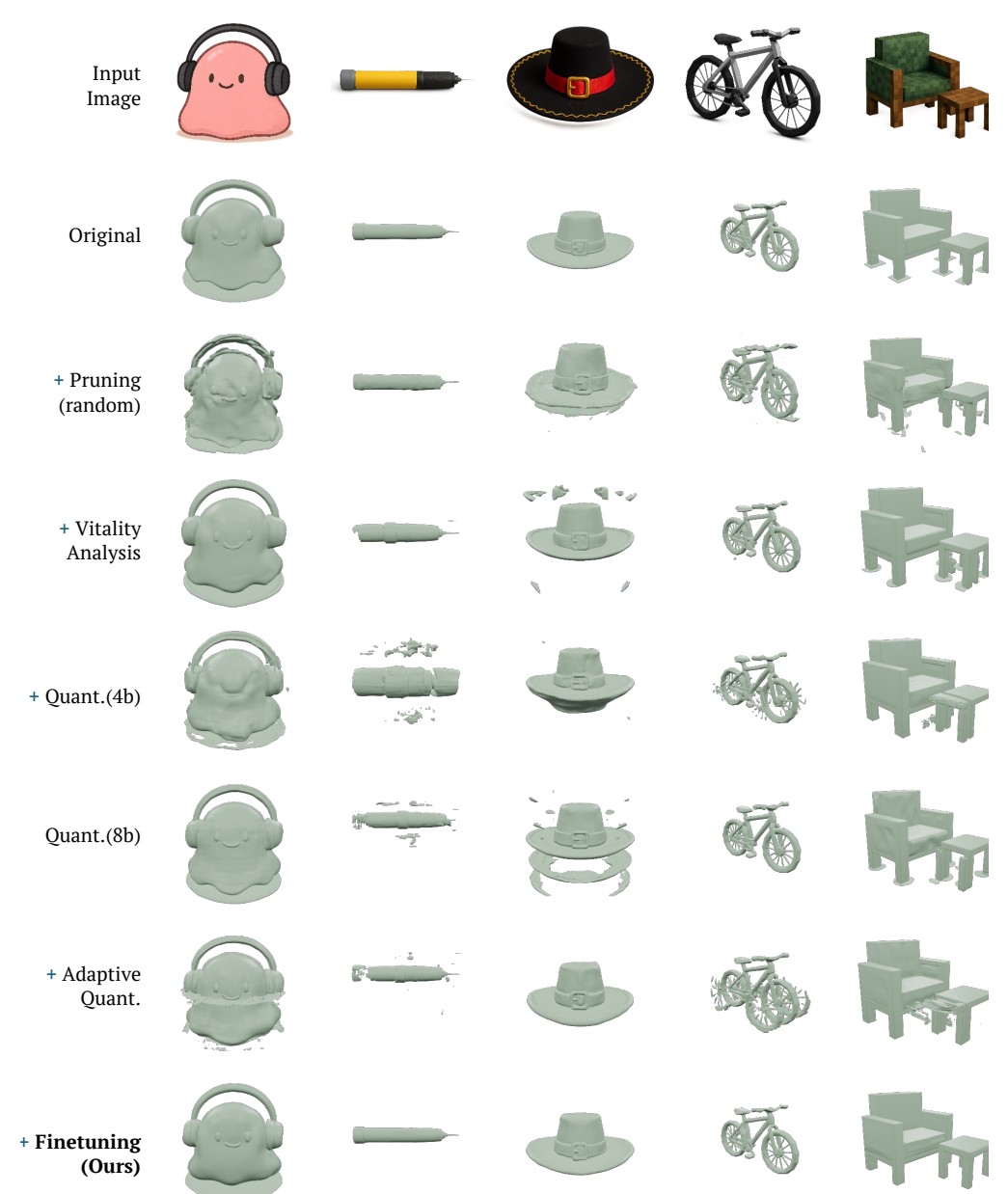

Figure D: **Additional Qualitative Ablation Results of Hunyuan3D 2.0.** Naive pruning and quantization introduce floaters and geometry collapse, while our method preserves quality nearly identical to the original.

structures, whereas quantizing all layers to 8-bit yields output quality that is nearly identical to the non-quantized model. In comparison to these models, our adaptive quantization strategy achieves a greater reduction in model size with substantially less degradation in performance. Despite these improvements, a residual discrepancy remains between the outputs of the vanilla model and ours, highlighting the necessity of the finetuning stage.

**Selection of Finetuning Strategies** To analyze the impact of different finetuning strategies, we conduct an ablation study on Hunyuan3D models (Li et al., 2025), comparing (i) full finetuning, (ii) selective finetuning applied only to the double- and single-block layers with the highest vitality scores (*i.e.*, "Max-vital" layers), and (iii) our proposed approach. Tab. D presents quantitative comparisons of different finetuning strategies on the Hunyuan3D models. We also provide qualitative ablations of the same models in Fig. G. We observe that training becomes unstable when all layers of the DiT

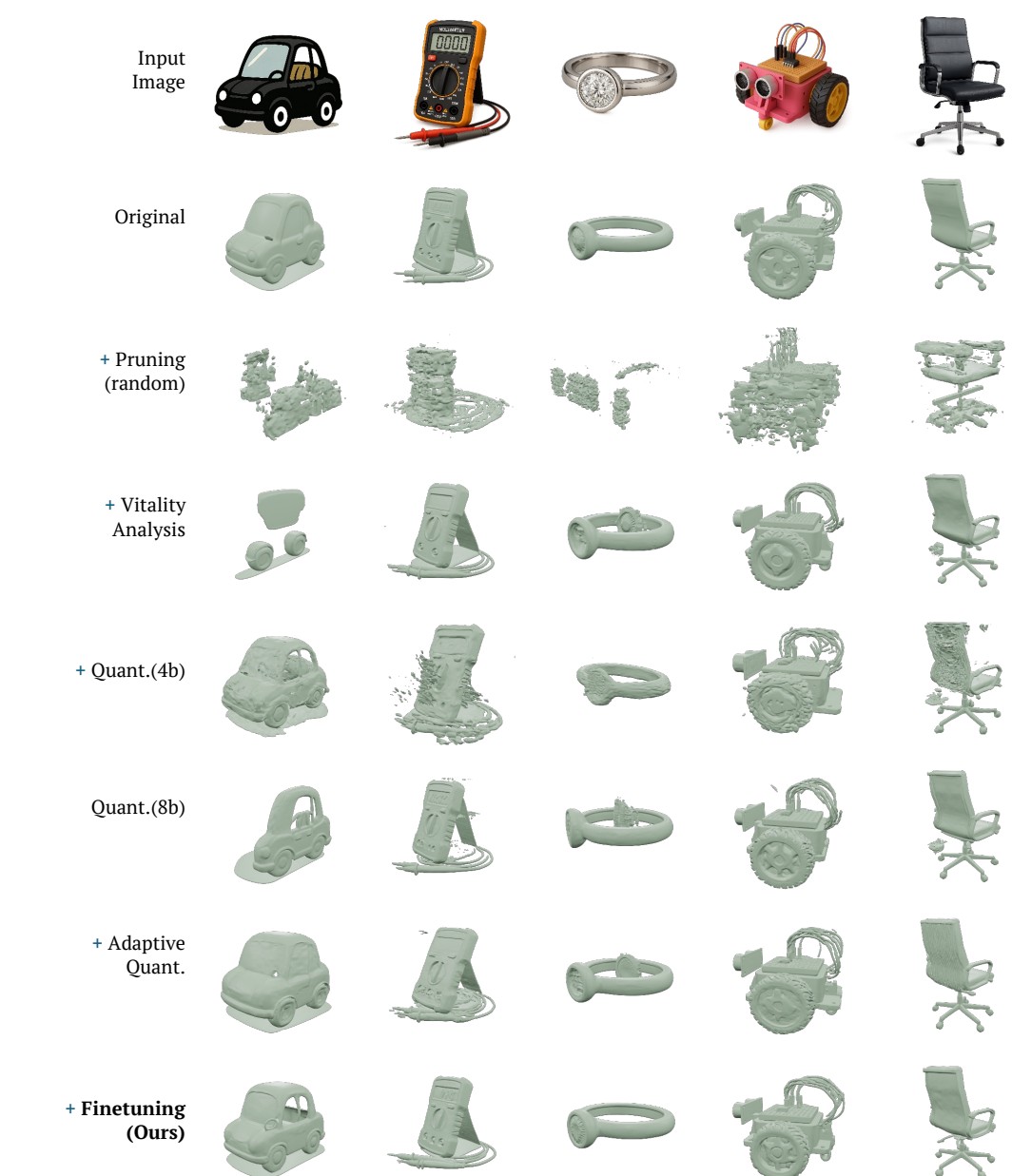

Figure E: **Additional Qualitative Ablation Results of Hunyuan3D 2mini.** Naive pruning and quantization introduce floaters and geometry collapse, while our method preserves quality nearly identical to the original.

architecture are finetuned simultaneously. Moreover, targeting only the "Max-vital" layers during finetuning often struggles to effectively mitigate degradation under compression, as it is difficult to recover finer details. To ensure both stability and effectiveness, our approach instead focuses on the "Min-vital" layers.

# E    DETAILED ANALYSIS OF VITALITY LAYERS

## E.1    ANALYSIS WITH CHAMFER DISTANCE METRICS

To support the proposed vitality score calculation method, we further show the vitality score analysis on different distance metrics in Fig. I. We show the analysis results of Chamfer distance. The

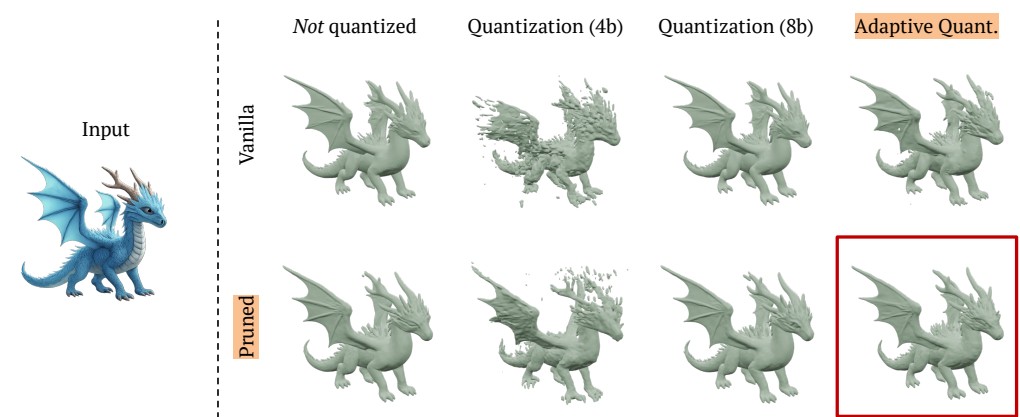

Figure F: **Component-wise Ablations Before Finetuning on Hunyuan3D 2.0.** Compared with other configurations, our approach (highlighted in the red box) effectively reduces memory cost with less degradation in generation quality. This remaining degradation indicates the need for an additional finetuning stage, as used in our method.

Table D: **Qualitative Comparison of Ablated Finetuning Strategies on Hunyuan3D Models** Our approach yields a more stable finetuning process than other strategies, improving overall shape quality.

| | Hunyuan3D 2.0 | | | | Hunyuan3D 2mini | | | |
|---|---|---|---|---|---|---|---|---|
| **Strategy** | Uni3D-I ↑ | OpenShape-I ↑ | V-IoU (%) ↑ | SS-IoU (%) ↑ | Uni3D-I ↑ | OpenShape-I ↑ | V-IoU (%) ↑ | SS-IoU (%) ↑ |
| Full-finetuning | 0.1766 | 0.0865 | 28.69 | 29.06 | 0.3210 | 0.1363 | 45.00 | 40.50 |
| w/ "Max-vital" | 0.3541 | 0.1490 | 61.50 | 56.68 | 0.3605 | 0.1479 | 66.93 | 62.28 |
| **Ours** | **0.3601** | **0.1491** | **72.04** | **68.31** | **0.3608** | **0.1484** | **73.77** | **70.36** |

quantitative analysis mostly follow the analysis result using EMD. Again, the analysis of Chamfer distance also show clear difference of layer contribution to output image. As shown in our analysis graph of Fig. J, we set non-vital layers as double block 7-11 and single block 7-23. In the qualitative analysis results, we can still observe that changes in vital layers (single 0-6 , double 0-6) produce significant deformation or degradation of detailed structure, while changes in non-vital layers do not make any major difference. The qualitative analysis again confirm our analysis results.

## E.2  ANALYSIS ON HUNYUAN3D MODELS

We also conduct a layer analysis on Hunyuan3D 2.0 using our vitality score computation method in Fig. K. Similar to Step1X-3D, we are able to distinguish between vital and non-vital layers; however, unlike Step1X-3D, where all layers beyond a certain index are non-vital, the Hunyuan model shows a mixed ordering of vital and non-vital layers. Moreover, the difference between vital and non-vital layers is less pronounced compared to Step1X-3D. This observation is also reflected in our ablation study: while Step1X-3D maintains performance close to the full model with layer pruning alone, the Hunyuan model exhibits slight artifacts without training. In case of qualitative analysis in Fig. L, modification of vital layers show severe deformation from original generated mesh as expected. When we remove non-vital layers which has small distance, the output meshes still show slight difference in high-frequency details.

For the Hunyuan3D 2mini model (Fig. M), which is already a compressed model with significantly fewer layers than the original, our layer analysis reveals that the number of layers with low vitality score (which can be regarded as non-vital) is fewer compared to larger-scale models. Consequently, the number of layers that can be pruned is more limited. Instead, we focus more on adaptive quantization with using used more 4-bit layers. In our qualitative analysis in FigureN, we can see that when removing the double layers, all the mesh outputs show geometric deformation from original meshes. In single block layers, we can also see there are some level of deformation in mesh details when removing front layers (0-13).

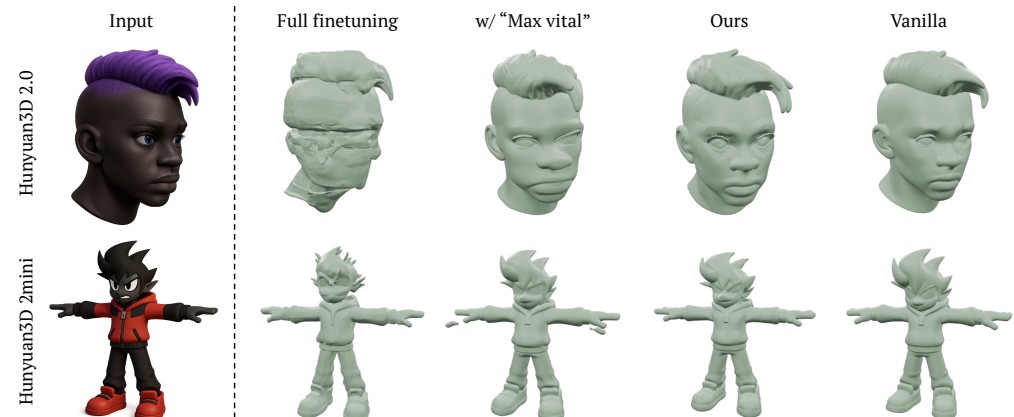

Figure G: **Qualitative Ablation of Finetuning Strategies on Hunyuan3D models.** The "Max-vital" layers denote those with the highest vitality (*i.e.*, contribute the most) per DiT block. We observe that fine-tuning only the lowest-vital ("Min-vital") layers leads to more stable learning.

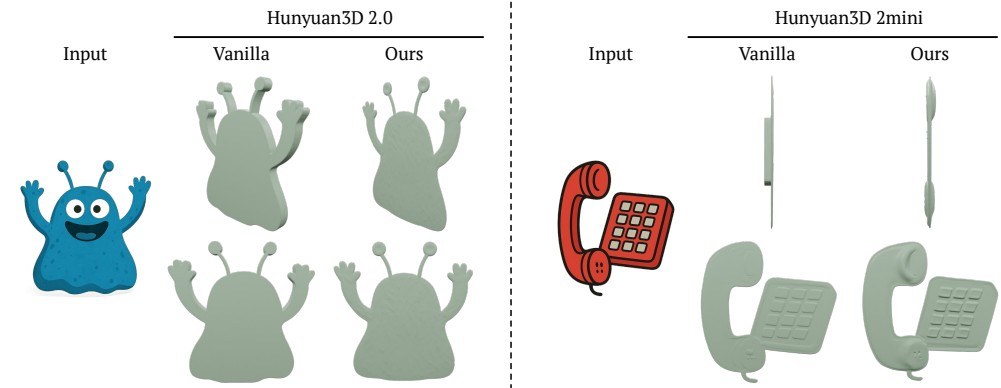

Figure H: **Limitations.** Since our approach aims for model compression while "maintaining" performance, the compressed model still shares core geometric limitations of the original framework.

## F LIMITATIONS AND FUTURE WORK

As mentioned in the main paper, our method successfully compresses 3D DiT models, achieving up to a 66% reduction in model size while maintaining nearly identical performance to the full-parameter model. While our quantization method supports precision down to 4 bits, we did not examine more extreme configurations (*e.g.*, 1-bit or 2-bit), which would require dedicated hardware-level implementations. Nevertheless, since our approach introduces general methodology for 3D generation model compression with layer-wise analysis, we expect it could be combined with hardware-level quantization research to achieve even greater compression efficiency.

Furthermore, our compressed framework does not overcome the core geometric and topological limitations inherent in the original model. As shown in Fig. H, Hunyuan-based models often fail to reconstruct accurate 3D structures from flat or stylized illustrations. Because our approach relies on distillation-based fine-tuning to match the original model's performance, these fundamental limitations are still preserved after compression.

As future work, we intend to further accelerate inference of the compressed model by reducing sampling steps and eliminating classifier-free guidance via knowledge distillation. In addition, we plan to extend our method to texture generation models, with the goal of building an efficient framework where both shape and texture generation are optimized. In parallel, since the current thresholds are manually tuned for each architecture, we plan to automate this process using relative vitality values across architectures. This will maintain the plug-and-play property while improving general applicability.

## G LLM USAGE

We utilized large language models (LLMs) exclusively for two purposes: (i) writing assistance and text refinement, including grammar checking and readability improvement, and (ii) generating input text descriptions required for the vitality layer analysis and evaluation. Specifically, for the second usage, we employed LLMs to sample text descriptions from Objaverse (Deitke et al., 2023) in a way that maximized category diversity while minimizing redundancy. Importantly, LLMs were not used for data analysis, interpretation, or generating any core research content.

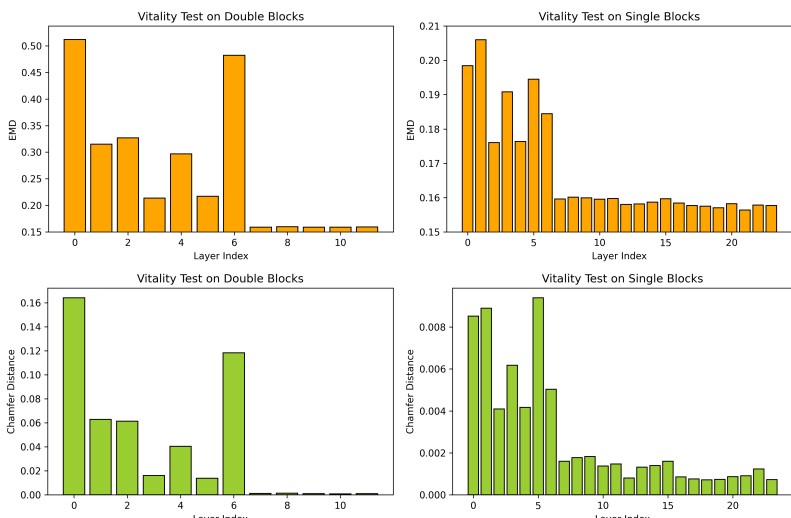

Figure I: **Detailed Vitality Analysis of Step1X-3D.** Up: Vitality analysis result with Earth Mover's Distance (EMD). Down : Analysis result with Chamfer Distance.

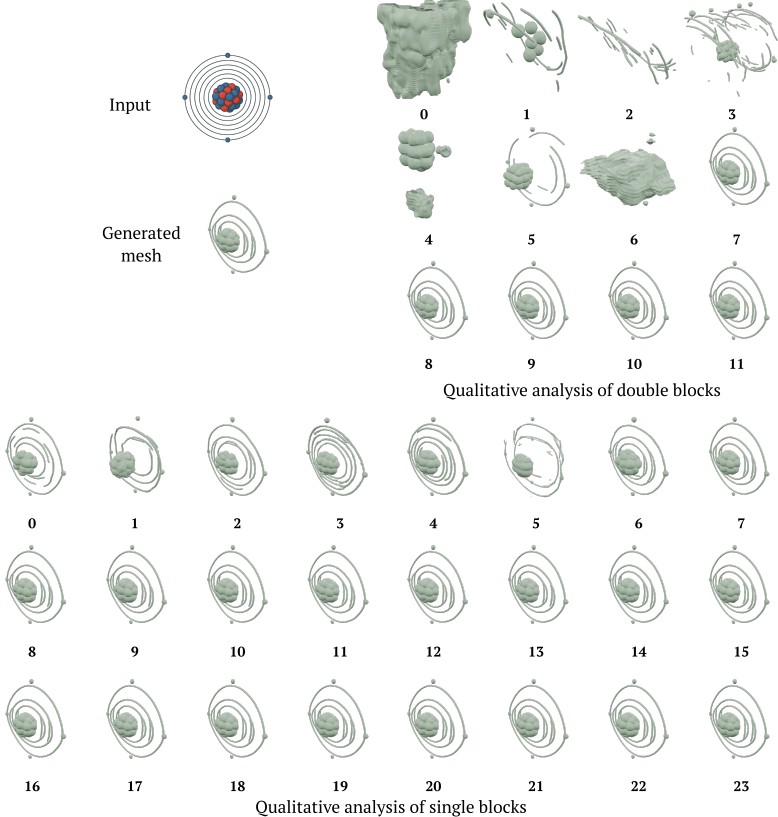

Figure J: **Meshes Generated After Layer Removal (Step1X-3D).** Numbers below each mesh denote removed layer indices. Removing double block layer 0–6 or single block layer 0–6 significantly degrades quality (vital layers), while removing other layers (non-vital) has minimal effect.

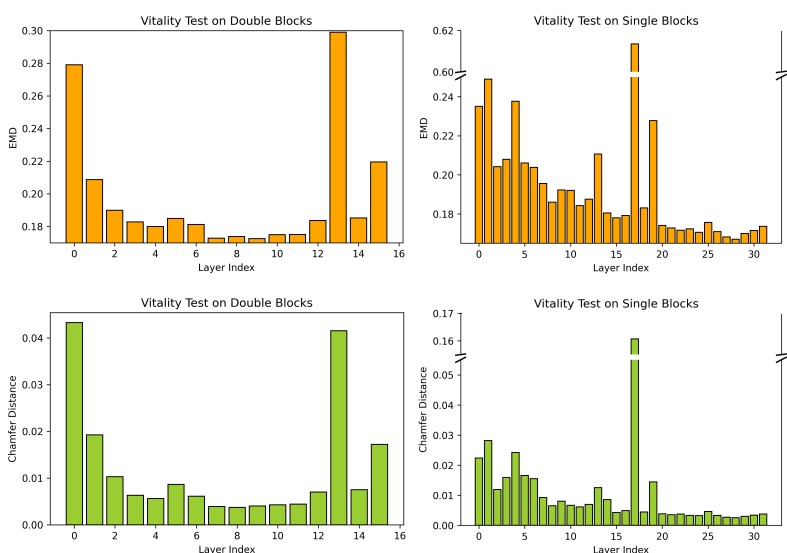

Figure K: **Detailed Vitality Analysis of Hunyuan3D 2.0.** Up: Vitality analysis result with Earth Mover's Distance (EMD). Down : Analysis result with Chamfer Distance.

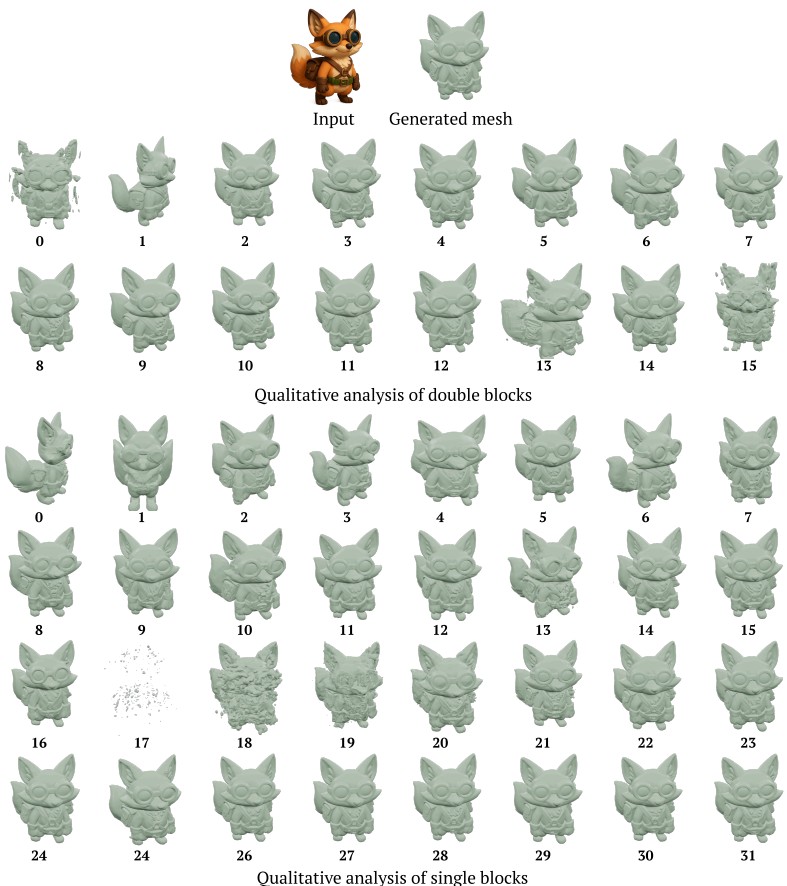

Figure L: **Meshes Generated After Layer Removal (Hunyuan3D 2.0).** Numbers below each mesh denote removed layer indices. Removing certain vital layers leads to severe quality degradation. Especially, removing single block layer 17 results in the complete collapse of the mesh.

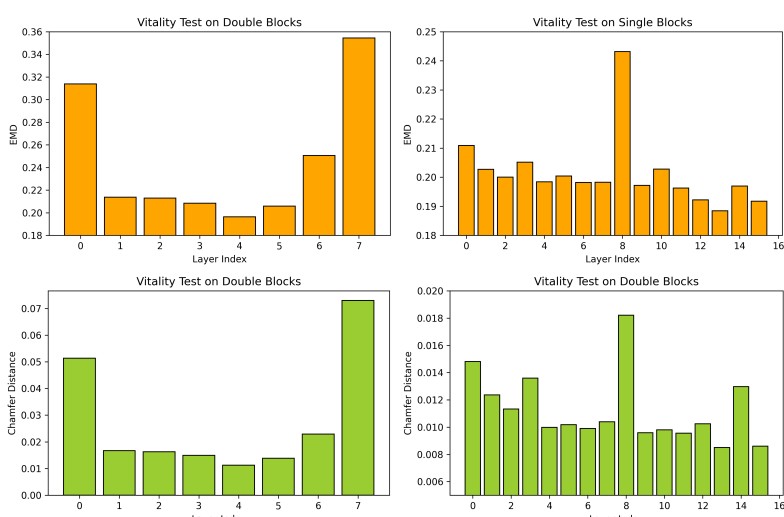

Figure M: **Detailed Vitality Analysis of Hunyuan3D 2Mini.** Up: Vitality analysis result with Earth Mover's Distance (EMD). Down : Analysis result with Chamfer Distance.

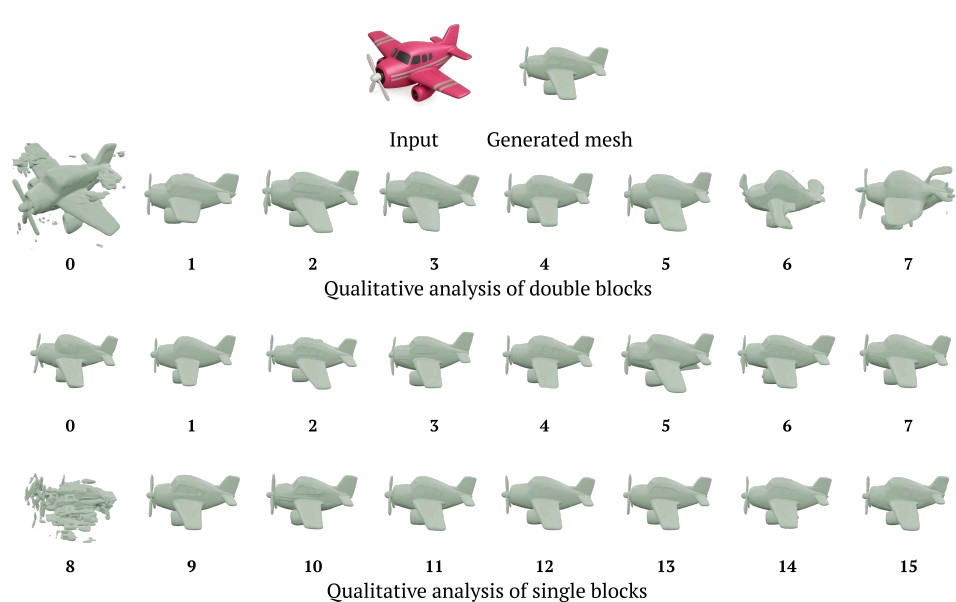

Figure N: **Meshes Generated After Layer Removal (Hunyuan3D 2mini).** Numbers below each mesh denote removed layer indices. Similar to Hunyuan3D 2.0, removing certain vital layers (single block layer 8) results in severe quality degradation. Despite being a lightweight variant, the model still contains non-vital layers whose removal has little impact on performance.

