# OpenReview forum: "Lightweight Image-to-3D Shape Generation via Vitality-Aware Pruning and Quantization"
_ICLR.cc/2026/Conference — Submitted to ICLR 2026_

### Official Review · Reviewer_i2aw · 2025-10-30

**Soundness:** 3
**Presentation:** 3
**Contribution:** 2
**Rating:** 6
**Confidence:** 3

**Summary:**

This paper introduces a compression framework for image-to-3D DiT models using layer vitality analysis, structured pruning, adaptive quantization, and targeted finetuning. Experiments on Step1X-3D, Hunyuan3D 2.0, and 2mini achieve 44-66% size reduction while maintaining synthesis quality.

**Strengths:**

1. This work addresses a practical problem with good results - 44-66% compression enables deployment in resource-constrained environments.
2. The evaluations are systematic and comprehensive, with results on different backbone models showing its generalization ability.
3. The idea of vitality analysis from T2I (Avrahami et al. 2025) to 3D generation using EMD on point clouds is a reasonable domain adaptation.

**Weaknesses:**

1. The technical novelty is relatively limited, as the core techniques (layer pruning, quantization, and distillation) are mostly from existing ones. The main contribution is applying vitality analysis (from Avrahami et al. 2025) to 3D with EMD. This is essentially applying a 2D technique to 3D generation with minimal domain-specific innovation. It would be better if there are some novel insights about 3D geometry or generation processes.
2. Different thresholds per architecture undermine "plug-and-play" claims. How sensitive are results? Can this be automated?
3. Why does selective finetuning of only the "Min-vital" layer work? There is no ablation study about this strategy.

**Questions:**

Refer to weakness.

---

> ### Author Response · Authors · 2025-11-25
>
> We sincerely thank the reviewer for their thoughtful and constructive feedback. For common concerns raised by multiple reviewers, we kindly refer the reviewer to our general comment for unified discussion.
>
> ## Threshold Automation
>
> We appreciate the reviewer’s insightful comment. The current thresholds are manually tuned for each architecture, but we plan to automate this process using relative vitality values across architectures. This will maintain the plug-and-play property while improving general applicability.
>
> ## Selective Finetuning Strategy
>
> Qualitative ablation results for the selective finetuning strategy are provided in the Appendix. These results demonstrate that updating the lowest-vitality (“Min vital”) layers effectively maintains generative quality while minimizing additional learning overhead.

---

> ### Author Response · Authors · 2025-11-28
>
> We kindly ask the reviewer to revisit our updated document together with our previous official comments. We provide ablated comparisons of finetuning strategies in Appendix D.2, Figure G, and we further describe our technical novelty in the common official comment. We sincerely appreciate the reviewer’s insightful suggestion regarding threshold automation, and we plan to explore this direction in future work.

---

### Official Review · Reviewer_UAxA · 2025-10-30

**Soundness:** 3
**Presentation:** 3
**Contribution:** 2
**Rating:** 4
**Confidence:** 4

**Summary:**

This paper has introduced the first study of prunning and quantization of I23D models. Specifically, by leveraging the structured pruning, vitality-aware adaptive quantization, and lightweight finetuning, fidelity can be well maintained under compression. Extensive experiments on several 3D base models and benchmarks have demonstrated the effectiveness of the proposed method.

**Strengths:**

1. With the rapid progress of 3D AIGC, the prunning and quantization of these models is indeed necessary.
2. The analysis and experiments of this paper is comprehensive.

**Weaknesses:**

1. I appreciate the The main concern is whether the proposed method is unique on 3D, or can be directly applied to any DiT/transformer architecture. Considering 3D generation all follow the same design, just applying existing prunning and quantization tricks on 3D DiTs can only be a weak contribution.
2. Some discussions of 3D generative models are missing, like 3DTopia-XL (CVPR 25), GaussianAnything (ICLR 25), EG3D / pi-GAN (GAN-based 3D generative models), and OpenAI's Shape-E (the first 3D diffusion model).

**Questions:**

1. I did not fully understand why the single-layer and double-layer block needs separate threshold and processing. Any insight behind?

---

> ### Author Response · Authors · 2025-11-25
>
> We sincerely thank the reviewer for their thoughtful and constructive feedback. For common concerns raised by multiple reviewers, we kindly refer the reviewer to our general comment for unified discussion.
>
> ## Additional References for Related Work
>
> We appreciate the reviewer’s helpful suggestions regarding additional references. We have incorporated the mentioned works into the *Related Work* section to provide a more comprehensive overview of recent 3D generative models.
>
> ## Threshold Designs
>
> We provide examples in the Appendix where applying the same thresholds to both double-layer and single-layer blocks causes significant structural distortion.

---

> ### Author Response · Authors · 2025-11-28
>
> We kindly ask the reviewer to revisit our updated document along with our previous official comments. We provide examples related to threshold designs in Appendix C, Figure A. We also emphasize our technical novelty in the common official comment. We also sincerely thank the reviewer’s suggestions and have updated the Related Work session.

---

### Official Review · Reviewer_Rc1K · 2025-11-01

**Soundness:** 3
**Presentation:** 3
**Contribution:** 1
**Rating:** 2
**Confidence:** 5

**Summary:**

This paper proposes a compression framework for image-to-3D generative models based on Diffusion Transformers (DiT). The core contribution is introducing "layer vitality" analysis to quantify each layer's contribution to generation quality using Earth Mover's Distance (EMD) between outputs of the full model and layer-ablated models. Based on this analysis, the method combines: Structured pruning of low-vitality layers, Vitality-aware adaptive quantization (8-bit for vital layers, 4-bit for others) and Selective finetuning targeting minimally-vital layers. Experiments on Step1X-3D, Hunyuan3D 2.0, and Hunyuan3D 2mini demonstrate effective model size reductions while maintaining synthesis quality comparable to full models.

**Strengths:**

1. Have analysis of layer importance in image-to-3D DiT models and first work to achieve actual model size reduction in this domain.
2. Achieves substantial compression (up to 66%) with minimal quality degradation, making high-quality 3D generation more accessible for resource-constrained environments.

**Weaknesses:**

1. Limited Novelty
This is essentially a straightforward application of existing compression techniques to 3D models, not a novel method:

"Layer vitality" via ablation+distance is directly borrowed from 2D image/video work (Avrahami 2025, Kim 2025)
Adaptive quantization based on importance scores is standard practice
The paper doesn't explain what makes 3D generation uniquely challenging for compression

2. No Theoretical Analysis
The paper is purely empirical without explaining why compression works:

Why do certain layers have low vitality? Is it due to training data, architecture design, or something else?
Why do different models (Step1X-3D vs. Hunyuan3D) show different vitality patterns?
Without understanding the underlying causes, it's unclear when this approach will succeed or fail

3. Missing Critical Experiments
Failure cases: No analysis of when/why compression degrades quality. What types of objects fail? At what compression ratio does quality collapse?
Computational cost: No reported training time, memory usage, or actual inference speedup. How expensive is the vitality analysis (requires N forward passes)? How long does finetuning take?
Baseline comparisons: No comparison with standard compression methods: Teacher-student distillation(e.g.,DMD) from scratch and other structured pruning approaches

4. Generalization Concerns

Different models need different hyperparameters (learning rates: 10⁻⁸ vs 10⁻⁴, different thresholds, different finetuning iterations)
Only 210 images used for vitality calculation—how stable are these scores?
Only tested on DiT models—unclear if it works for other 3D architectures such as AR-based models.

5. Insufficient Ablations

How does each component (pruning, quantization, finetuning) contribute individually?
How sensitive is performance to threshold selection

**Questions:**

Why not try sparse-voxel-based methods such as Trellis? What are the challenges?
How sensitive is the method to the number of samples used for vitality calculation?
Since 3D generation models are not particularly large, speedup seems less critical to me. Why not focus on general model compression task or video generation instead, which is more crucial?

---

> ### Author Response · Authors · 2025-11-25
>
> We sincerely appreciate the reviewer's detailed and thoughtful feedback. We value the time and effort dedicated to carefully analyzing our work and raising valuable questions. The reviewer's comments have helped us clarify our technical contributions. For common concerns raised by multiple reviewers, we kindly refer the reviewer to our general comment for unified discussion.
>
> ## Details of Our Main Technical Contribution
> 3D diffusion-based generative models are inherently black-box systems, integrating geometry, view consistency, and spatial reasoning under complex training conditions. This makes theoretical analysis highly challenging, as layer contributions depend on both architecture design and training objectives (e.g., view-aligned consistency, geometry priors). Hence, our work emphasizes an “discovery” of layer vitality patterns, which itself novel insight into how 3D DiTs allocate computation across layers.
>
> Moreover, our approach introduces a simple yet effective compression pipeline that does not require additional data-driven training processes. This design choice enhances the practical applicability and scalability of compression for 3D shape generative models.
>
> ## Limitations
>
> Since our finetuning stage primarily focuses on mitigating performance degradation during compression through teacher–student distillation, the compressed framework still inherits the geometric and topological limitations of the original model. Example cases are provided in the Appendix.
>
> ## Methodological Details
>
> ### Structural Pruning and Distillation-Based Finetuning
> Our method indeed employs a teacher-student distillation strategy during finetuning, yet it does not rely on DMD. Our goal is to demonstrate that memory consumption can be reduced without significant performance degradation by using the most standard and widely adopted compression approach. Furthermore, we have focused on a structured pruning scheme, ensuring that the model compression is performed in a principled manner. We believe that integrating more advanced distillation techniques such as DMD could further enhance the results. We plan to explore this direction in future work.
>
> ### Threshold Selection
> We provide qualitative results for different thresholds in adaptive quantization in the appendix. Since the original model primarily consists of weights represented in FP16, applying 4-bit or 8-bit quantization inevitably leads to information loss, which can affect the performance of a more lightweight framework.
>
> ### Robustness to the Number of Sampling Points on Vitality Analysis
> Additional results on the effect of the number of samples used in computing per-layer vitality are provided in the Appendix.
>
> ## Generalized Concerns
> Our framework is designed for DiT-based 3D generative models, which share a common denoising-based generation process. This makes them particularly suitable for layer-wise vitality analysis, as in prior work, and for applying structured compression based on the resulting insights.
>
> Other architectures, such as AR-based models, would require a different treatment of temporal or autoregressive dependencies, which lies beyond the current scope but remains an interesting direction for future work.
>
> ## Insufficient Ablations
>
> We provide the effect of threshold selection in layer pruning and adaptive quantization in the Appendix. Meanwhile, conducting separate experiments for each individual component would require substantial computational resources. We believe our ablation study already provides sufficient evidence of each contribution, and we have further supplemented it by including the missing ablations of the finetuning stage in the Appendix.
>
> ## Clarification on Research Scope and Contribution
>
> We appreciate the reviewer’s suggestion to consider broader generative settings such as general model compression or video generation. However, we specifically target 3D DiT-based generative models, where geometry synthesis and denoising-based generation introduce fundamentally different computational and representational challenges compared to 2D or temporal domains.
>
> While extending our approach to other modalities (e.g., video or general generative models) would indeed be valuable, such directions require handling distinct spatiotemporal dependencies and objectives, which are beyond the scope of this study. We believe that this effective compression framework for 3D DiT models provides a strong foundation for the future generalization to broader domains.

---

> ### Author Response · Authors · 2025-11-28
>
> We kindly ask the reviewer to revisit our updated document together with our previous official comment. In that comment, we provide detailed responses regarding our main technical contribution, methodological details, generalization concerns, and clarification of our research scope and contributions. We additionally include evaluations addressing the limitations (Appendix F, Figure H) and further methodological details such as threshold selection in adaptive quantization (Appendix C.3, Figure B) and the robustness of metrics for vitality analysis (Appendix C.1, Table A). We also address data sufficiency for our analysis and evaluation, as well as the technical novelty of our work, in the common official comment.

---

### Official Review · Reviewer_ts7t · 2025-11-01

**Soundness:** 3
**Presentation:** 3
**Contribution:** 2
**Rating:** 4
**Confidence:** 4

**Summary:**

The paper proposes a compression pipeline for large image-to-3D DiT models. The central idea is to estimate each layer’s *vitality* by measuring the 3D output degradation via Earth Mover’s Distance between point clouds after temporarily removing that layer. Layers with low vitality are pruned, while remaining ones are quantized with adaptive bit widths proportional to vitality. A small distillation fine-tuning step attempts to recover performance. Experiments on Step1X-3D, Hunyuan3D 2.0, and Hunyuan3D 2-mini report parameter reduction with minimal drop in embedding-based metrics (Uni3D-I, OpenShape-I).

**Strengths:**

**Clear motivation.**
3D DiT models are large, so compression is valuable.

**Practical meanings.**
The parameter reduction without catastrophic quality loss is practical.

**Good writing.**
The core idea of this paper is clearly written and explained in the method section. The ablation study cleanly separates pruning, quantization, and finetuning effects, and main comparison experiments are clearly demonstrated.

**Weaknesses:**

**Marginal novelty.**
The approach reuses well-known pruning and quantization frameworks from 2D diffusion literature, adding no new theoretical formulation, and its contribution mainly lies in empirical replication on 3D models. It only substitutes the 2D similarity metric with a 3D EMD and apply the same concept to 3D DiTs. The quantization and finetuning steps are standard, and the final pipeline is an incremental adaptation rather than a new principle.

**Computational cost and scalability not addressed.**
The vitality analysis itself appears computationally heavy. The paper does not disclose GPU hours, runtime, or scaling properties. Without these, claims of *lightweight* or *efficient* are debatable and the significance of this approach would suffer.

**Insufficient evaluation.**
All quantitative metrics are embedding-based and no direct 3D geometry comparisons are provided, which leaves uncertainty about how much detail or structural accuracy is lost during pruning. The reported metrics are calculated from only 200 test samples, which is too small to support broad claims and hinders credibility.

**Questions:**

1. What is the actual computational cost of vitality analysis?

2. Is there a particular reason why the reported metrics can only be evaluated with 200 pairs?

3. Can you report Chamfer Distance or other geometry-level metrics on a small benchmark?

4. What are the real inference-time savings (latency and memory) after compression?

---

> ### Author Response · Authors · 2025-11-25
>
> We sincerely thank the reviewer for their thoughtful and constructive feedback. For common concerns raised by multiple reviewers, we kindly refer the reviewer to our general comment for unified discussion.
>
> ## Actual Computational Cost for Vitality Analysis
>
> Vitality analysis requires running an additional inference-like pass per layer to measure each layer’s contribution. Therefore, its computational cost can be approximated as ``inference time × (1 + N_layers) × N_images'', which depends on both the model’s size and its original inference efficiency.
>
> ## Geometry-Level Metrics
>
> We provide both volumetric (V-IoU) and systemic surface IoU (SS-IoU) metrics in the Appendix to evaluate geometric consistency throughout the compression process of each model.

---

> > ### Author Response · Authors · 2025-11-28
> >
> > We kindly ask the reviewer to revisit our updated document along with our previous official comments. In the comment, we provide details on how to estimate the actual computational cost for vitality analysis, and we report geometry-level metrics in Appendix D.2, Table B. We also address data sufficiency for our analysis and evaluation in the common official comment, and we include a quantitative comparison of inference cost savings in Appendix D.2, Table C.

---

### Author Response · Authors · 2025-11-25

We sincerely thank the reviewers’ thoughtful feedback and valuable suggestions. Their comments have helped us improve the clarity and rigor of our work.

We have carefully revised the paper by incorporating additional experiments, technical clarifications, and an updated introduction.
We kindly encourage the reviewers to refer to the revised version for these updates, which we believe further strengthen the motivation and contributions of our work.

Below, we address the main concerns and clarify our contributions regarding technical novelty, efficiency insights, and dataset preparation for analysis and evaluation.

## Technical Novelty

Our work presents the **first** approach for compressing image-to-3D shape generative models, introducing a vitality-driven framework that directly reduces computational redundancy in 3D DiT architectures.

As discussed in the introduction, recent studies on efficient 3D generation primarily focus on accelerating inference, rather than reducing spatial computation to improve memory efficiency. Inspired by recent analyses that measure per-layer contributions within the DiT architecture, we extend this concept to the 3D DiT framework, which has not been previously explored in the context of model compression.

In contrast to prior methods that exploit contribution analysis to enhance editability during generation, our method leverages **vitality-aware** metrics to **systematically identify** and **remove redundant layers or unimportant computations** that contribute less to the synthesis process.

## VRAM Allocation and Inference Efficiency.

As suggested by the reviewer *ts7t* and *Rc1K*, we report both VRAM usage and inference time comparisons in the Appendix,.

Since our approach includes a layer pruning step that progressively reduces model size, it naturally improves inference efficiency in both time and memory consumption.
Although our main objective is physical model compression rather than inference acceleration, the resulting model exhibits gains in computational efficiency in practice.

## Data Sufficiency for Analysis and Evaluation

Our experimental settings, including the number of samples used for analysis and evaluation, follow those established in prior work.

For example, Stable Flow [1] employs only 64 text prompts for analysis, while TV-LiVE [2] examines layer vitality patterns using 40 text prompts. Similarly, Step1X-3D [3] utilizes 110 images for evaluation, and Craftman3D [4] uses 30 scenes randomly selected from the GSO dataset for validation. Following these precedents, we use 210 images for vitality analysis and 200 images for performance evaluation.

To ensure class diversity, we first curated 1,000 text prompts representing diverse object categories using GPT (LLM) based on the Objaverse dataset, and subsequently generated corresponding images with DALL$\cdot$E 3, as mentioned in the paper.

**References**

[1] Avraham, Omri et al., Stable Flow: Vital Layers for Training-Free Image Editing, CVPR 2025.

[2] Kim, Min-Jung et al., TV-LiVE: Training-Free, Text-Guided Video Editing via Layer Informed Vitality Exploitation, arXiv 2025.

[3] Li, Weiyu et al., Step1X-3D: Towards High-Fidelity and Controllable Generation of Textured 3D Assets, arXiv 2025.

[4] Li, Weiyu et al., CraftsMan3D: High-fidelity Mesh Generation with 3D Native Generation and Interactive Geometry Refiner, ICLR 2024.

---

### Author Response · Authors · 2025-11-29

Dear Reviewers,

We recently received a notice from the conference committee, which you have likely seen as well, stating that the discussion period has been temporarily halted due to the incident that occurred yesterday.

While this unexpected pause is unfortunate, we want to reassure you that we remain fully committed to engaging with all of your thoughtful comments and continuing to strengthen our work.

For the rest of the rebuttal period, we will continue improving the paper based on your suggestions and feedback.
We will make every effort to address your concerns thoroughly and to present our contribution as clearly and usefully as possible for the community.

Thank you again for your engagement and valuable insights.
We appreciate your patience and hope you will stay tuned for our future updates.

Warm regards,

The Authors

---

### Author Response · Authors · 2025-12-04

We thank the reviewers for their constructive feedback.\
In response to the requests, we have updated our submission and highlight the following clarifications and newly added results:

**Component-wise Ablation Before Finetuning (Appendix D.2, Figure F)** *by reviewer Rc1K, related to Insufficient Ablations*

We additionally provide ablation studies separating the effects of layer pruning and quantization prior to finetuning. These results allow a clearer understanding of the individual contributions of each component in our compression pipeline.


**Quantitative Comparison of Selective Finetuning Strategies on Hunyuan3D 2.0 (Appendix D.2, Table D.)** *by Reviewer i2aw*

We also include quantitative comparisons of different selective finetuning strategies applied to the Hunyuan3D 2.0 models. The results demonstrate that our proposed method yields improved stability during distillation compared to other strategies.
Since we expanded our ablations in the appendix, the numbering of some figures and tables has changed.

To avoid confusion in the discussion, we refer to the indices used in the reviewers’ original comments when citing relevant figures/tables.\
Please refer to our _**updated paper**_ for the newly added ablations.

We hope these additions address the concerns and help clarify the contributions of our work.\
We sincerely appreciate your time and consideration.

---

### Meta-Review · Area_Chair_Daha · 2026-01-05

**Summary:**

The reviewers raised concerns regarding the technical novelty, the validity of the proposed vitality metric, and the overall strength of the empirical evidence. In particular, a key concern is whether the proposed layer vitality estimation provides a reliable and principled signal for pruning in generative models.
Several reviewers also questioned whether the method meaningfully accounts for inter-layer dependencies and whether the reported gains go beyond straightforward applications of existing pruning, quantization, and fine-tuning techniques. These concerns collectively informed the recommendation.

**Reviewer Concerns:**

Concerns addressed by the rebuttal:
The rebuttal provided additional ablations and clarifications on the effects of individual components (e.g., pruning, quantization, and finetuning), which partially address concerns about implementation details and experimental transparency.

Existing concerns:
However, the core concerns regarding the validity of the proposed vitality metric remain unresolved. The paper defines layer importance by removing a single layer and measuring output differences using EMD, but the choice of EMD is not sufficiently justified over alternative metrics, and it primarily evaluates geometric discrepancies while ignoring other important aspects of generative quality such as texture and appearance.
More fundamentally, the vitality analysis is performed in a layer-wise and isolated manner, without accounting for strong correlations and interactions between layers in deep generative models. As a result, observing that a single layer appears unimportant in isolation does not imply that it remains unimportant when combined with other layers.

These unresolved issues undermine the reliability of the pruning criterion and limit the strength of the paper’s conclusions.

**Reviewer Scores:**

Reviewer ts7t: Likely to maintain their original score, as the main concerns regarding limited novelty, evaluation metrics, and scalability were only partially addressed and do not affect the core validity issues.

Reviewer Rc1K: Unlikely to change their score. Their concerns about limited novelty, lack of principled analysis, and missing baselines remain largely unaddressed.

Reviewer UAxA: Might slightly decrease or maintain their score after discussion, given that the rebuttal does not resolve whether the proposed method offers 3D-specific insights beyond direct adaptations of existing techniques.

Reviewer i2aw: Unlikely to increase their score. While the rebuttal adds some ablations, the central questions about the validity of the vitality metric and the necessity of the proposed pipeline remain open.

---

### Decision · Program_Chairs · 2026-01-26

Reject